# A FAK/HDAC5 signaling axis controls osteocyte mechanotransduction

Tadatoshi Sato[1], Shiv Verma[1], Christian D. Castro Andrade[1], Maureen Omeara[1], Nia Campbell[1], Jialiang S. Wang[1], Murat Cetinbas[2], Audrey Lang[1], Brandon J. Ausk[3], Daniel J. Brooks[1,4], Ruslan I. Sadreyev[2], Henry M. Kronenberg[1], David Lagares [5], Yuhei Uda [6], Paola Divieti Pajevic[6], Mary L. Bouxsein [1,4], Ted S. Gross[3] & Marc N. Wein [1,7✉]

Osteocytes, cells ensconced within mineralized bone matrix, are the primary skeletal mechanosensors. Osteocytes sense mechanical cues by changes in fluid flow shear stress (FFSS) across their dendritic projections. Loading-induced reductions of osteocytic Sclerostin (encoded by *Sost*) expression stimulates new bone formation. However, the molecular steps linking mechanotransduction and *Sost* suppression remain unknown. Here, we report that class IIa histone deacetylases (HDAC4 and HDAC5) are required for loading-induced *Sost* suppression and bone formation. FFSS signaling drives class IIa HDAC nuclear translocation through a signaling pathway involving direct HDAC5 tyrosine 642 phosphorylation by focal adhesion kinase (FAK), a HDAC5 post-translational modification that controls its subcellular localization. Osteocyte cell adhesion supports FAK tyrosine phosphorylation, and FFSS triggers FAK dephosphorylation. Pharmacologic FAK catalytic inhibition reduces *Sost* mRNA expression in vitro and in vivo. These studies demonstrate a role for HDAC5 as a transducer of matrix-derived cues to regulate cell type-specific gene expression.

[1] Endocrine Unit, Massachusetts General Hospital, Harvard Medical School, Boston, MA, USA. [2] Department of Molecular Biology and Department of Pathology, Massachusetts General Hospital, Harvard Medical School, Boston, MA, USA. [3] Department of Orthopaedics and Sports Medicine, University of Washington, Seattle, WA, USA. [4] Center for Advanced Orthopaedic Studies, Department of Orthopedic Surgery, Beth Israel Deaconess Medical Center, Harvard Medical School, Boston, MA, USA. [5] Center for Immunology and Inflammatory Diseases, Fibrosis Research Center, Massachusetts General Hospital, Harvard Medical School, Boston, MA, USA. [6] Translational Dental Medicine, Henry M. Goldman School of Dental Medicine, Boston University, Boston, MA, USA. [7] Broad Institute of Harvard and MIT, Cambridge, MA, USA. ✉email: mnwein@mgh.harvard.edu

The skeleton constantly adapts to changes in mechanical cues. Exercise stimulates new bone formation while skeletal unloading, as experienced by astronauts during prolonged space flight, causes accelerated bone loss[1]. Osteocytes, postmitotic cells of the osteoblast lineage buried deeply within bone matrix, are the primary mechanosensors in bone[2–4]. The molecular mechanisms through which osteocytes transduce mechanical inputs are poorly understood.

Osteocytes control bone remodeling by secreting paracrine factors that in turn regulate activity of osteoclasts and osteoblasts on bone surfaces[5]. Of these osteocyte-derived paracrine factors, RANKL and Sclerostin (encoded by the Sost gene) are both central regulators of bone remodeling. Osteocyte-derived RANKL is a crucial osteoclastogenic factor[6], and the target of the osteoporosis drug denosumab[7]. Sclerostin is a canonical WNT pathway inhibitor that blocks osteoblast activity stimulated by WNTs[8]. Romosozumab, a neutralizing sclerostin antibody, is now approved for osteoporosis treatment[9,10]. Sost expression by osteocytes is mechanically regulated, with sclerostin levels increasing with unloading[11] and decreasing with skeletal loading[12]. Osteocytic Sost downregulation is important for loading-induced bone formation[13], and Sost upregulation contributes to immobilization-induced bone loss[14,15].

While it is clear that modulating Sost expression is an important strategy used by osteocytes to link mechanical cues to bone formation, the intracellular signaling pathways through which this occurs are largely unknown. Like mechanical loading, parathyroid hormone (PTH) stimulates bone formation, in part, by reducing sclerostin levels[16,17]. Sost expression is positively regulated by the transcription factor MEF2C, which binds to a + 45 kB downstream enhancer site[18,19] that is absent in high bone-mass patients with Van Buchem disease[20]. In many biologic systems, class IIa histone deacetylases are potent inhibitors of MEF2-driven gene expression[21]. Class IIa HDACs are uniquely endowed with long N-terminal extensions that confer responsiveness to external signals and allow inhibitory binding to MEF2 family transcription factors[22]. HDAC4 and HDAC5 inhibit MEF2-driven osteocytic Sost expression[23]. Moreover, PTH signaling drives HDAC4/5 translocation from the cytosol to the nucleus via a cAMP-dependent pathway involving inhibition of salt-inducible kinases[24]. Despite these advances, whether class IIa HDACs participate in osteocyte mechanotransduction and loading-induced Sost suppression is currently unknown.

It is generally accepted that osteocytes sense mechanical cues by changes in fluid-flow shear stress (FFSS) across their dendritic processes[25,26]. Skeletal loading induced during functional activity primarily places long bones in bending[27], which due to heterogeneous strain distribution within a given cross-section facilitates interstitial fluid flow within the lacunar–canalicular system[28,29]. This interstitial FFSS produces focal strains at attachment sites surrounding osteocyte cell processes[30]. Integrin αV/ß3 heterodimers have been proposed to play a key role in osteocyte/matrix interaction and mechanotransduction[31–33]. Multiple membrane proximal signaling mechanisms have been described downstream of FFSS across dendritic processes. These include outside-in integrin signaling, ATP release[34], local calcium fluxes[35], TRPV4-mediated microtubule reorganization and ROS generation[36], plasma membrane disruptions[37], and effects on connexin hemichannels[38]. However, precise links between these proximal signaling steps and Sost suppression remain to be determined.

Here, we report that FFSS triggers class IIa HDAC nuclear translocation in osteocytes, and that HDAC4/5 are required for loading-induced bone formation in vivo. While class IIa HDACs are involved in both PTH and FFSS-mediated Sost suppression, these two external cues utilize distinct upstream signaling mechanisms to drive HDAC4/5 nuclear translocation. In osteocytes, constitutive cell/matrix interactions lead to basal activation of focal adhesion kinase (FAK) through outside-in integrin signaling[39] for review of integrin-mediated signaling). FAK is known to play crucial roles in mechanotransduction in many tissue types[40–43], although links between FAK and class IIa HDACs have not been described. Here, we show that FAK regulates class IIa HDAC subcellular localization by direct HDAC5 tyrosine 642 phosphorylation. FFSS inhibits FAK activity, a step that is required for FFSS-induced Sost suppression. Moreover, many of the transcriptomic effects of FFSS are mimicked by small molecule FAK inhibitors, and by RGD peptides that block integrin/matrix adhesion. Finally, pharmacologic FAK inhibitors can suppress Sost expression in vivo, indicating the therapeutic potential of this FAK/class IIa HDAC/Sost signaling axis.

## Results

**Mechanosensitive class IIa HDACs are required for loading-induced bone formation.** We previously demonstrated that parathyroid hormone (PTH) signaling promotes the dephosphorylation and nuclear translocation of HDAC4 and HDAC5 in osteocytes, and that HDAC4/5 are required for PTH-induced suppression of Sost expression in vitro and in vivo[24]. Mechanical cues and PTH signaling both suppress Sost expression and stimulate new bone formation. However, the precise signaling mechanisms used by mechanical loading to reduce Sost expression remain unknown. Here, we asked if HDAC4/5 are required for mechanical loading-induced bone formation. For this, right tibiae of mice were subjected to a 3-week osteogenic cantilever bending loading regimen (see Supplementary Fig. 1 for an overview of this model and the relationship between strain distribution and periosteal response)[44]. Cortical bone morphology was obtained via μ-CT imaging of the tibia mid-shaft prior to the experiment, and beam theory was used to define load magnitudes such that peak-induced normal strains were comparable across all mutant strains. This loading regimen leads to significant gains in periosteal bone formation in control mice, global $Hdac5^{-/-}$ mice, and mice lacking $Hdac4$ in osteoblasts/osteocytes (DMP1-Cre; $Hdac4^{f/f}$, hereafter referred to as $Hdac4^{ob/ocy}$). In contrast, $Hdac4^{ob/ocy};Hdac5^{-/-}$ double mutants (hereafter referred to as $Hdac4/5^{DKO}$) fail to increase periosteal bone formation following this loading regimen (Fig. 1a).

Since we previously reported that $Hdac4/5$ mutant mice show osteopenia[24], we also performed biomechanical testing of the femoral diaphysis in control and $Hdac4/5$ compound mutants. As shown in Supplementary Fig. 2, bone biomechanical properties in $Hdac4/5^{DKO}$ are lower than control, due to reduced bone size and cortical mass. The normal relationship between bone mass and strength is preserved, however, suggesting no deficits in bone quality in $Hdac4/5^{DKO}$ animals.

Next, we determined the role of class IIa HDACs in loading-induced sclerostin suppression. While this cyclical-loading regimen reduces sclerostin immunoreactivity and gene expression in control mice, this does not occur in $Hdac4/5^{DKO}$ animals (Fig. 1b–d). Sclerostin is a potent inhibitor of the WNT signaling pathway[8]. Sclerostin downregulation in osteocytes is needed for loading-induced WNT pathway activation and osteoblastic bone formation[13,45]. Sclerostin suppression and subsequent osteocytic WNT pathway activation occurs in a complex manner related to patterns of strain experienced by individual osteocytes[46]. Control mice show robust increases in periosteal cells staining positive for the activated form of ß-catenin; in contrast, this periosteal accumulation of WNT-responsive cells is absent in $Hdac4/5^{DKO}$ mice (Fig. 1e, f).

Although bone cells can respond to multiple potential types of mechanical cues, it is widely accepted that fluid-flow shear stress

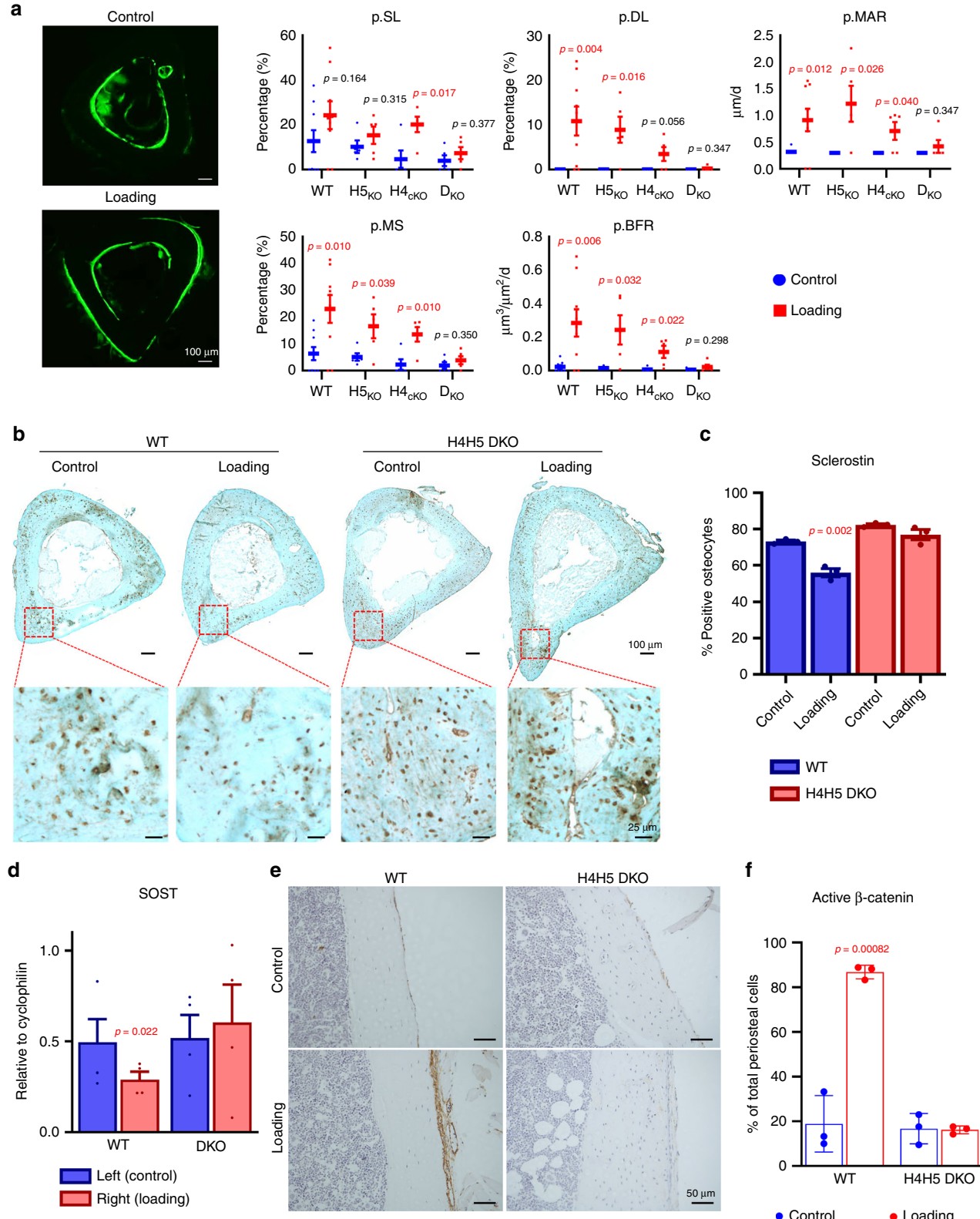

(FFSS) across osteocytic dendritic processes is a major mechanism used for mechanotransduction[3]. To complement these in vivo loss of function studies, we exposed control and HDAC4/5-deficient Ocy454 cells (a murine mechanosensitive osteocytic cell line[47]) to FFSS. These experiments reveal two key differences between WT and HDAC4/5-deficient cells. First, basal *Sost* mRNA (in static conditions) is significantly increased in the absence of HDAC4/5. Second, HDAC4/5 are required for FFSS-induced *Sost* suppression in vitro (Fig. 2a). Taken together, these results demonstrate that HDAC4/5 are required for loading-induced bone formation and sclerostin suppression in vivo and in vitro.

Having established a crucial role for class IIa HDACs in responses to mechanical cues in vivo, we next asked how

**Fig. 1 Class IIa HDACs are required for loading-induced bone formation. a** WT, HDAC4-cKO (*HDAC4^fl/fl^;DMP1-cre*), HDAC5-KO, and H4H5-DKO mice were subjected in vivo cantilever bending of the right tibia. Each mouse underwent a 3-week regimen (3 days/week, 100 cycles/day, 2500 με peak normal strain), and dynamic histomorphometry was performed on the tibia mid-shaft. Calcein labeling was performed at 2 days and 11 days prior to sacrifice. Exogenous loading significantly increased p.BFR in WT, HDAC4-cKO, and HDAC5 KO compared with contralateral tibiae. No significant p.BFR elevation observed in H4H5-DKO mice compared with contralateral tibiae (*n* = 5–9 mice per group). p.SL (periosteal single-labeling surface), p.DL (periosteal double-labeling surface), pMAR (periosteal mineral apposition rate), p.MS (periosteal mineralizing surface), p.BFR (periosteal bone-mineral formation rate), WT (wild-type mice), H5KO (HDAC5^−/−^), H4KO (HDAC4^fl/fl^;DMP1-cre), DKO (H4H5-DKO, HDAC5^−/−^; HDAC4^fl/fl^;DMP1-cre). **b** Sclerostin immunohistochemistry (IHC) was performed in WT and H4H5-DKO mice (*n* = 3). High-magnification images show representative images of sclerostin-positive and -negative cells in cortical bones. **c** All transverse sections were counted by ImageJ. Sclerostin-positive cells numbers are normalized by entire osteocyte number. (*n* = 3 mice per group) *P*-values vs control (contralateral tibia). **d** qRT-PCR analyses from bone marrow-flushed tibias of WT and H4H5-DKO mice. Exogenous loading significantly reduced *Sost* mRNA expression in WT, but not H4H5-DKO mice. (*n* = 4) *P*-values vs contralateral tibia. **e**, **f** Non-phospho (active) β-catenin IHC in WT and H4H5-DKO mice. Exogenous loading increased active β-catenin staining in periosteal cells of WT, but not in H4H5-DKO mice. Each experiment was repeated three times. (*n* = 3 mice per group) *P*-values vs control (contralateral tibia) are shown in the figure. Two-sided unpaired *t* test was used (**a, c, d, f**). Data are expressed as mean ± SEM. Source data are provided as a Source Data file.

HDAC4/5 participate in osteocyte mechanotransduction. PTH signaling drives class IIa HDAC dephosphorylation and nuclear translocation[24]. In a similar manner, HDAC4/5 translocate from the cytoplasm to the nucleus in response to FFSS in Ocy454 cells (Fig. 2b–d). Class IIa HDAC nuclear translocation is often accompanied by reduced phosphorylation at serines in the N-terminal domain of the proteins[48]. Only modest reductions in HDAC4/5 N-terminal serine phosphorylation are observed after FFSS exposure (Supplementary Fig. 3A), suggesting that class IIa HDAC localization may be controlled by FFSS signaling via a mechanism independent of N-terminal serine phosphorylation. Supporting this notion, FFSS further suppresses *Sost* expression in cells expressing the HDAC5 S259A/S498A variant[49] (Supplementary Fig. 3B, C).

**Focal adhesion kinase orchestrates mechanosensitive *Sost* expression in osteocytes**. We next addressed the upstream mechanisms linking FFSS signaling and changes in osteocyte gene expression. Transcriptomic analysis of Ocy454 cells after 3 h exposure to FFSS revealed a core group of 369 regulated transcripts (Fig. 2e; Supplementary Dataset 1 for all RNA-seq data). Next, we explored the functional role of HDAC4/5 in FFSS-induced gene expression changes in Ocy454 cells. For this, control and HDAC4/5-deficient (DKO) cells were treated plus/minus FFSS for 3 h followed by RNA-seq. HDAC4/5 DKO cells show intact FFSS-induced p42/44 ERK phosphorylation (Supplementary Fig. 3D); however, these cells fail to regulate the majority FFSS-dependent transcripts (Fig. 2f).

We next sought clues to provide a link between FFSS signaling and HDAC4/5 nuclear translocation in osteocytes. Gene ontology and upstream pathway analysis[50] of the FFSS-dependent differentially expressed genes suggested that focal adhesion may play an important role in the coordinated gene expression changes that were observed (Fig. 2g, h). Therefore, we asked what effect focal adhesion kinase (FAK) inhibitors (such as PF562271[51]) might have on basal and FFSS-regulated *Sost* expression. In parallel, pharmacologic agents that target other candidate mechanosensitive signaling pathways were tested (UO126 is a MEK inhibitor[52], PD98059 is a ERK inhibitor[53], Y-27632 is a ROCK inhibitor[54], SB431542 is a TGFβR inhibitor[55], CCG1423 is a MRTF inhibitor[56], verteporfin is a YAP inhibitor[57]). Indeed, pre-treating Ocy454 cells with FAK inhibitors (PF562271, VS-6063, PF431396) reduced basal *Sost* expression without cytotoxicity (Fig. 3a) and blocked further FFSS-induced suppression of *Sost* and regulation of other mechanosensitive transcripts (Supplementary Fig. 4A). Consistent with its effects on blocking FFSS-induced *Sost* suppression, PF562271 pretreatment also blocked FFSS-induced HDAC4/5 nuclear translocation (Supplementary Fig. 4B, C). Taken together,

the fact that FAK emerged from upstream pathway analysis and studies with inhibitors of candidate mechanotransduction pathways prompted us to focus in more detail on the role of FAK in FFSS-induced *Sost* suppression in osteocytes.

Next, we generated FAK-deficient Ocy454 cells to confirm results seen with pharmacologic FAK inhibitors. As expected, FAK-deficient cells have reduced phosphorylation of the canonical FAK substrate Paxillin at Y118 (Fig. 3b for single-cell FAK-mutant clones and Supplementary Fig. 5A, B for the results with bulk CRISPR/Cas9-mutated cells). Similar to what is observed with PF562271 treatment, FAK-deficient cells show two significant abnormalities at the level of *Sost* expression. First, FAK deficiency significantly reduces basal *Sost* expression. Second, cells lacking FAK fail to regulate *Sost* and the mechanosensitive target gene *Dmp1*[58] in response to FFSS, but not PTH, treatment (Fig. 3c, d; Supplementary Fig. 5C, D). In addition, FAK inhibitors do not block PTH-induced *Sost* suppression (Supplementary Fig. 5E). Cells lacking FAK show normal (mild) reductions in HDAC4/5 N-terminal serine phosphorylation in response to prolonged FFSS treatment (Supplementary Fig. 6A), again suggesting that FFSS/FAK-mediated SOST suppression occurs via a pathway independent of HDAC4/5 serine phosphorylation. To determine the global role of FAK in gene expression changes in response to mechanotransduction, control and FAK-mutant single-cell clones were exposed to FFSS followed by RNA-seq analysis. FAK-deficient cells show normal FFSS-induced p42 ERK phosphorylation (Supplementary Fig. 3D); however, FAK-deficient cells fail to regulate the vast majority of FFSS-sensitive transcripts (Fig. 3e–g). Taken together, these pharmacologic and genetic results support a key role for FAK in mechanosensitive gene expression changes in osteocytes.

**FFSS signaling suppresses FAK activity, and FAK inhibitors mimic many effects of FFSS**. Having demonstrated that FAK is crucial for osteocytes to respond to FFSS, we next determined the effects of acute FFSS exposure on FAK activity. Surprisingly, Ocy454 cells (which synthesize large amounts of extracellular matrix[23,47]) show high basal FAK activity when grown under static conditions. FFSS treatment rapidly reduces FAK Y397 autophosphorylation and, consequently, Paxillin Y118 phosphorylation (Fig. 4a). Since FFSS reduces FAK activity, we asked if small-molecule FAK inhibitors might mimic the effects of FFSS. As expected, treating Ocy454 cells with PF562271 or VS-6063[59] leads to dose-dependent reductions in FAK Y397 autophosphorylation (Fig. 4b). Consistent with our model, FAK inhibitors suppress *Sost* expression in cells grown in static conditions (Fig. 3a) in a HDAC4/5-dependent manner (Fig. 4c). Therefore, the presence of FAK is necessary for FFSS to suppress *Sost*, and

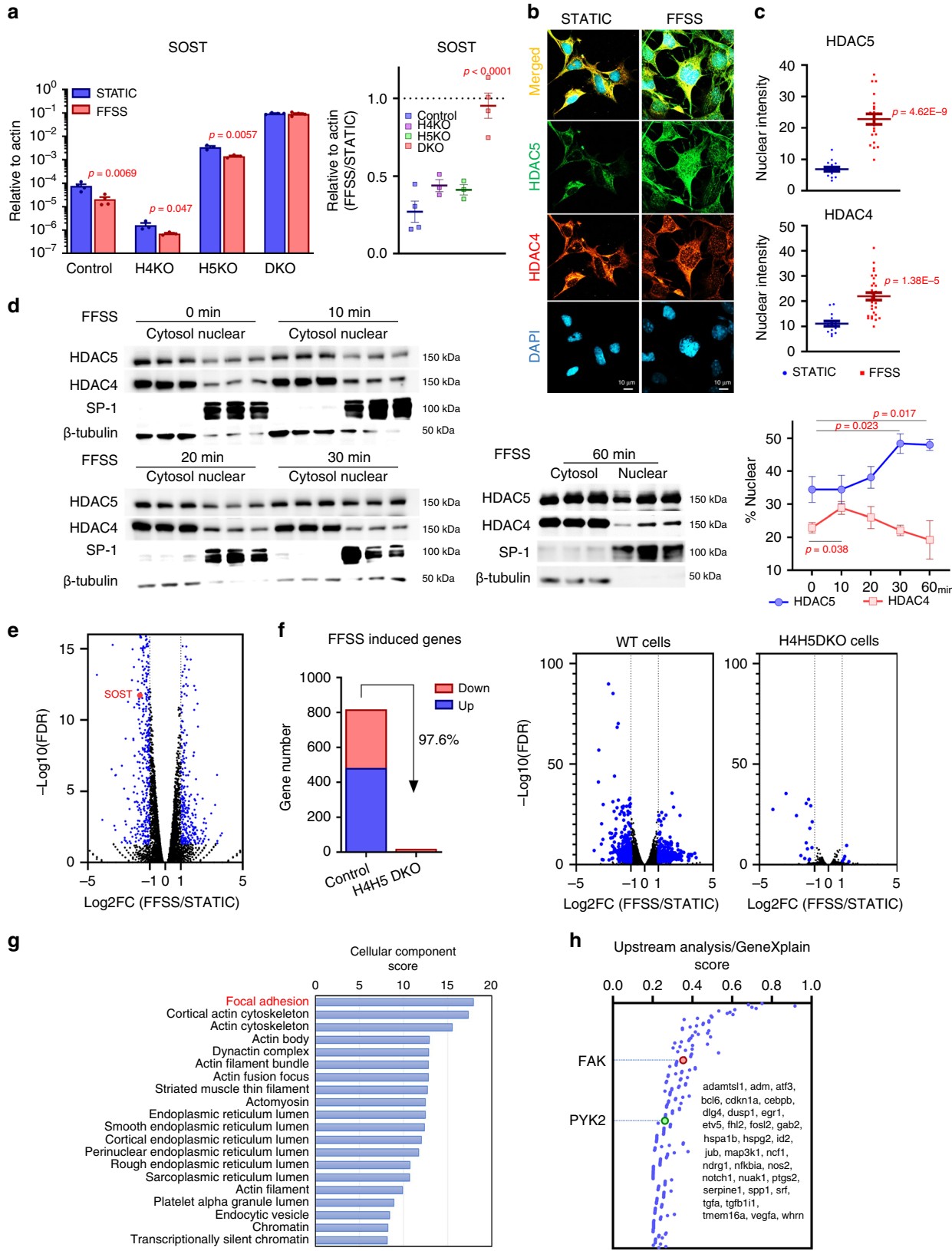

FAK inhibition is sufficient to reduce *Sost* even in the absence of FFSS.

Next, we performed RNA-seq to define the global transcriptomic effects of the FAK inhibitor PF562271 in Ocy454 cells (Fig. 4d, 975 genes were regulated >twofold with FDR < 0.05). Consistent with previous RT-qPCR data, *Sost* was detected as a

significantly downregulated transcript in this analysis. Combining inter-related RNA-seq data sets (Fig. 4e), we note that each manipulation (FFSS, FAK deletion, and FAK inhibitor treatment) leads to distinct but overlapping patterns of gene expression changes. First, 200 of the 975 genes regulated by PF562271 were also regulated by FAK gene deletion (*P* < 4.987e-13 for the

**Fig. 2 Fluid-flow shear stress (FFSS) regulates Sost expression through HDAC4/5 nuclear translocation and changes in FAK/integrin signaling. a** WT, HDAC4 KO, HDAC5-KO, and HDAC4/HDAC5 (H4H5) DKO Ocy454 cells were treated plus/minus FFSS for 3 h, followed by RT-qPCR for *Sost*. FFSS for 3 h reduced *Sost* expression in WT cells to a greater degree than in H4H5-DKO cells. *P*-values vs STATIC condition. Absolute SOST expression data are shown in the left panel ($n = 4$ cells per group for control and DKO, $n = 3$ cells per group for H4KO and H5KO). The right panel shows the ratio of SOST comparing FFSS versus static-treatment within each cell line. *P*-values adjusted for multiple comparisons (versus control) are shown in the right panel. **b**, **c** HDAC5-deficient Ocy454 cells infected with FLAG-tagged HDAC5 were subjected to FFSS (3 h) followed by immunocytochemistry staining for HDAC4/5 and then confocal microscopy. Pictures show a representative image of each condition. HDAC5 (green, localization determined by anti-FLAG immunostaining), endogenous HDAC4 (red), and DAPI (blue). The nuclear densitometric intensity of FLAG-HDAC5 and endogenous HDAC4 were measured by ImageJ. *P*-values vs STATIC condition ($n = 15$ cells for STATIC, $n = 21$ cells for HDAC5 FFSS, $n = 29$ cells for HDAC4 FFSS). Each experiment was repeated three times. **d** Ocy454 cells were subjected to FFSS for the indicated times and subjected to subcellular fractionation followed by immunoblotting. The nuclear fraction of endogenous HDAC5 and HDAC4 was quantified by densitometry over time. ($n = 3$ biologic cell replicates were performed) *P*-values for each time point versus 0 min are shown in the right panel. **e** Volcano plot showing up- and downregulated genes by RNA-seq by FFSS treatment for 3 h in Ocy454 cells. Blue dots represent significantly regulated genes (log2 fold change (FFSS/STATIC) < −1 or >1, FDR < 0.05), black dots represent genes whose expression is not significantly regulated by FFSS. *Sost* gene was confirmed as a downregulated gene. **f** Control and HDAC4/5-deficient Ocy454 cells were treated plus/minus FFSS for 3 h followed by RNA-seq. Volcano plots as in **e** are shown. The majority of FFSS-induced DEGs are not regulated in cells lacking HDAC4/5. **g** Gene ontology analysis of FFSS-regulated genes showed enrichment of pathways linked to integrin/FAK signaling (matrix cellular components). In this graph, the *x* axis corresponds to the gene ontology enrichment score (adjusted *P*-value) for genes differentially expressed in response to FFSS. **h** Upstream analysis suggests that FAK and PYK2 are potential upstream candidate regulators of the coordinated gene expression changes seen by RNA-seq in response to FFSS. In this graph, the *x* axis corresponds to the upstream analysis score (adjusted *P*-value) for genes that were differentially expressed in response to FFSS. Listed in the figure are the FFSS-induced DEGs found in the upstream FAK-dependent "signature". One-sided **a**, **d** and two-sided **c** unpaired *t* test, and one-way analysis of variance (ANOVA) followed by Tukey–Kramer post hoc test (**a** in the right panel) were used. Data are expressed as mean ± SEM. Source data are provided as a Source Data file.

statistical significance of the overlap between two groups), indicating largely concordant effects between acute pharmacologic and chronic genetic FAK inhibition. Notably, 83 of the 362 genes (including *Sost*) regulated by FFSS are also regulated by PF562271 treatment ($P < 1.480e-20$). Furthermore, 210 of the 362 genes regulated by FFSS were also regulated by FAK deletion ($P < 9.643e-97$). Therefore, this analysis defines a core (nonrandom) set of mechanosensitive transcripts that are FAK-dependent. That being said, many (111/362, 30.7%) mechanosensitive genes are regulated by neither FAK deletion nor PF562271, indicating that additional FAK-independent mechanotransduction signaling arms exist in osteocytes. In addition, expression of many genes regulated by PF562271 or FAK gene deletion are not regulated by FFSS (Fig. 4e, f), indicating that FAK likely controls cellular functions independent of its role in FFSS signaling. Finally, we cannot rule out the possibility that some basal gene expression changes in response to lentiviral infection per se may occur. However, all RNA-seq studies used cells infected (or not) under identical experimental conditions for side-by-side comparison. Furthermore, principle component analysis of all 28 RNA-seq libraries analyzed here demonstrates only modest effects of lentiviral infection at the transcriptomic level (Supplementary Fig. 6B). In sum, these results indicate that FFSS signaling in osteocytes inhibits FAK activity, and that FFSS-mediated FAK inhibition is a critical step in downstream regulation of *Sost* and many additional mechanosensitive target genes.

**Fluid-flow shear stress regulates FAK-mediated class IIa HDAC tyrosine phosphorylation.** Next, we sought to understand the link between FFSS-induced FAK inhibition and class IIa HDACs. While serine/threonine phosphorylation is known to regulate class IIa HDAC subcellular localization[22], tyrosine phosphorylation of these proteins has not been studied to date. Phosphotyrosine immunoprecipitation leads to recovery of endogenous HDAC4 and HDAC5 in Ocy454 cells, and rapid (10 min) FFSS treatment reduces this association (Fig. 5a). Furthermore, a specific phosphotyrosine immunoreactive species at the expected HDAC5 size (140 kDa) is recovered by FLAG-HDAC5 immunoprecipitation, and is reduced by FFSS (Fig. 5b).

Similarly, FLAG-HDAC4 immunoprecipitates with phosphotyrosine antibodies in a manner that was enhanced by vanadate and reduced by PF562271 treatment (Fig. 5c). Since both FAK inhibitors and FFSS (which inhibits FAK) both reduce class IIa HDAC tyrosine phosphorylation, these results suggest that class IIa HDACs are tyrosine-phosphorylated in cells in a FAK-dependent manner.

Recombinant proteins were then used to see if FAK could directly phosphorylate class IIa HDAC proteins. First, FAK-dependent HDAC4 and HDAC5 phosphorylation was observed in bioluminescent-based kinase assays (Fig. 5d; Supplementary Fig. 7A, B, E4Y1 polymers are used as a positive control FAK substrate, and FAK inhibitors, as expected, block FAK-mediated HDAC5 phosphorylation). To directly visualize FAK-dependent class IIa HDAC phosphorylation, $\gamma^{32}$P-ATP kinase assays were performed which demonstrate FAK-mediated HDAC5 phosphorylation (Fig. 5e). To complement these studies, in vitro kinase assays were analyzed by phosphotyrosine immunoblotting (Fig. 5f), which again show direct FAK-dependent HDAC5 phosphorylation in vitro.

Finally, recombinant full-length HDAC5 (expressed in Sf9 cells) phosphorylated by FAK in vitro was subjected to chymotryptic digestion followed by mass spectrometry to identify direct phosphorylation site(s). No tyrosine-phosphorylated sites were observed in recombinant HDAC5 in the absence of FAK treatment (Fig. 6a). In contrast, FAK-induced HDAC5 tyrosine phosphorylation is only detected at residue Y642 (Fig. 6b; Supplementary Fig. 7C, Ascore value for FAK-dependent HDAC5 Y642 phosphorylation is 35.0), a consensus FAK phosphorylation site[60]. FAK-dependent phosphotyrosine content was significantly reduced in a HDAC5 Y642F mutant compared with WT HDAC5 (Supplementary Fig. 7D, E). Next, we generated and immuno-purified antisera that recognizes HDAC5 phosphorylated at Y642 (Fig. 6c; Supplementary Fig. 7F; this antisera does not recognize HDAC5 Y642F, and no immunoreactive bands are noted in HDAC5-deficient cells). In osteocytes, we note that HDAC5 Y642 phosphorylation is reduced by FAK inhibitors, FAK gene deletion, and FFSS achieved both by rotating platform and laminar flow chambers (8 dynes/cm$^2$) (Fig. 6d; Supplementary Fig. 7G, H). To test the functional role of HDAC5 Y642

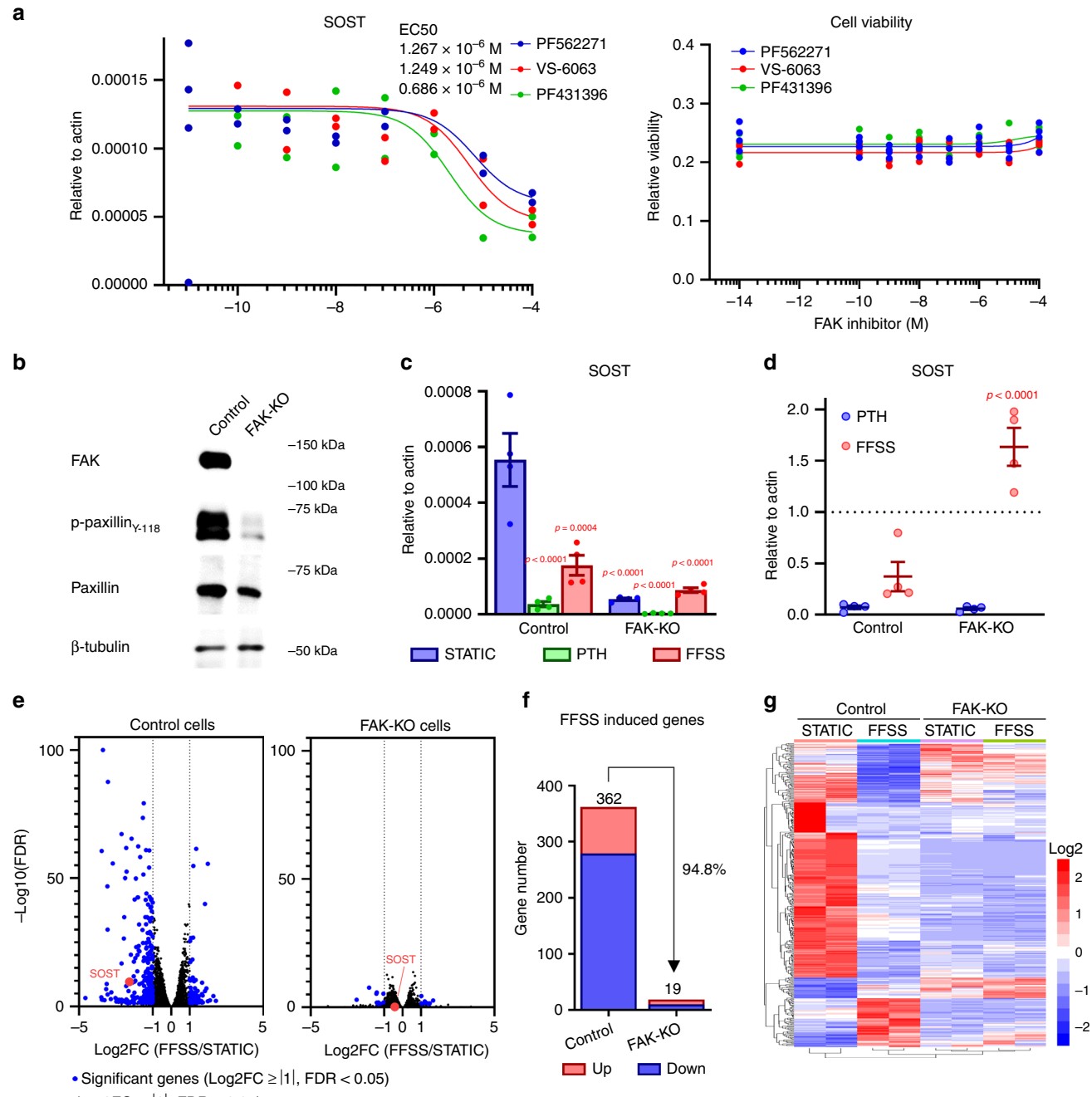

**Fig. 3 FAK is required for FFSS-induced gene expression changes in osteocytes. a** Left, *Sost* mRNA expression was assessed after treatment with FAK inhibitors (PF562271, VS-6063, and PF431396) for 4 h (*n* = 2 biologic replicates were performed). All FAK inhibitor treatments reduced *Sost* mRNA expression in Ocy454 cells grown in static conditions. Right, cell viability was measured after 4 h treatment with the indicated FAK inhibitors (*n* = 4 biologic cell replicates were performed). Data are shown with each measured value and best fit curve. **b** Single-cell FAK-KO cells were generated by CRISPR-Cas9. Protein expression of FAK, p-paxillin (Y118), and paxillin in control and FAK gene deleted (FAK-KO) Ocy454 cells. Phospho-paxillin was significantly decreased in FAK-KO cells. Each experiment was independently repeated three times. **c, d** Control and FAK-KO cells were as indicated for 3 h followed by RT-qPCR analysis for *Sost*. For PTH treatments, cells were treated under static condition at a PTH 1–34 concentration of 242 nM. While basal Sost levels are reduced in FAK-KO cells, there is no further decrease in response to FFSS. In contrast, PTH further suppresses SOST in FAK-deficient cells. (*n* = 4 biologic cell replicates were performed for RNA analysis) *P*-values adjusted for multiple comparisons are shown versus STATIC condition in control cells. **e** RNA-seq analyses were performed with control and FAK-KO cells treated for 3 h with/without FFSS. Volcano plots representing the effects of FFSS in each cell type (control and FAK-KO) are shown. Differentially expressed genes (DEGs, log2FC >1 or < −1, FDR < 0.05) are shown as blue points. While many DEGs are noted in response to FFSS in control cells, very few FFSS-induced DEGs are detected in FAK-KO cells. **f** Genes whose expression was significantly changed by FFSS in control and FAK-KO cells. **g** Most FFSS-regulated gene expression changes require the presence of FAK. Heatmap of expression values for DEGs, shown as *Z* scores of log2(CPM) values for a given gene across all samples. Rows, genes; columns, samples of control and FAK-KO single-cell clones. One-way ANOVA followed by Tukey–Kramer post hoc test was used (**c, d**). Data are expressed as mean ± SEM. Source data are provided as a Source Data file.

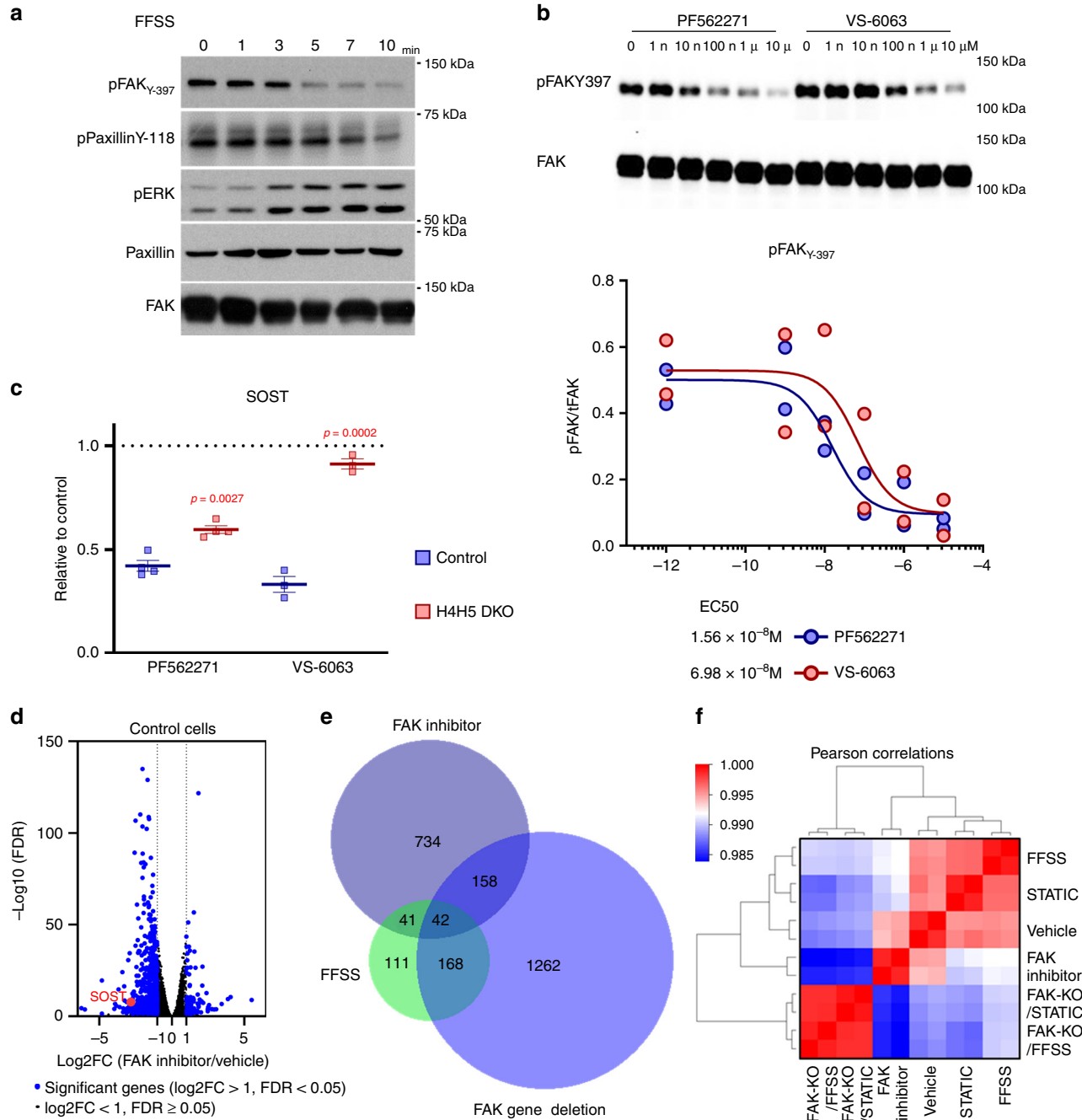

**Fig. 4 FFSS reduces FAK activity, and FAK inhibitors mimic effect of FFSS. a** Protein expression of phospho-FAK (Y397), p-paxillin (Y118), and paxillin in Ocy454 cells exposed to FFSS for the indicated times. Phospho-FAK (Y397) was significantly decreased by FFSS within 10 min, and paxillin (the substrate of FAK) phosphorylation was decreased at later time points. **b** Dose responses of FAK inhibitors (PF562271 and VS-6063) on Ocy454 cells. FAK inhibitors were treated with indicated concentrations for 1 h. FAK inhibitors significantly reduced phospho-FAK (y-397) activity. Immunoblotting data are quantified in the bottom graph ($n = 2$ biologic protein replicates were performed). Data are shown with each measured value and best fit curve. **c** Control and HDAC4/5 compound mutant (H4H5-DKO) cells were treated plus/minus FAK inhibitors for 4 h followed by SOST RT-qPCR. H4H5-DKO cells fail to suppress SOST in response to FAK inhibitor treatment. ($n = 4$ RNA replicates for control, and $n = 3$ RNA replicates for H4H5-DKO were performed) $P$-values adjusted for multiple comparisons vs control are shown. **d** A volcano plot of RNA-seq analysis with FAK inhibitor, PF562271. RNA-seq analysis was performed with Ocy454 cells by vehicle and PF562271 treatment for 4 h. Significant genes were determined as log2FC > |1| and FDR < 0.05. **e** A Venn diagram of significantly changed genes among FFSS, FAK inhibitor (PF562271), and FAK gene deletion. FFSS was performed for 3 h with Ocy454 cells infected with a mock sgRNA-expressing lentivirus. Control uninfected Ocy454 cell were treated with FAK inhibitor for 4 h. FAK gene deletion was carried out by CRISPR-Cas9, and single-cell FAK-KO cell line was generated from bulk FAK-KO cell lines with puromycin selection. **f** A Pearson correlation matrix heatmap of RNA-seq analysis. Ocy454 cells were treated with/without FFSS, with/without FAK inhibitor (PF562271), and with/without FAK gene deletion by CRISPR-Cas9. One-way ANOVA followed by Tukey–Kramer post hoc test were used (**c**). Data are expressed as mean ± SEM. Each experiment was repeated three times (**a**, **b**). Source data are provided as a Source Data file.

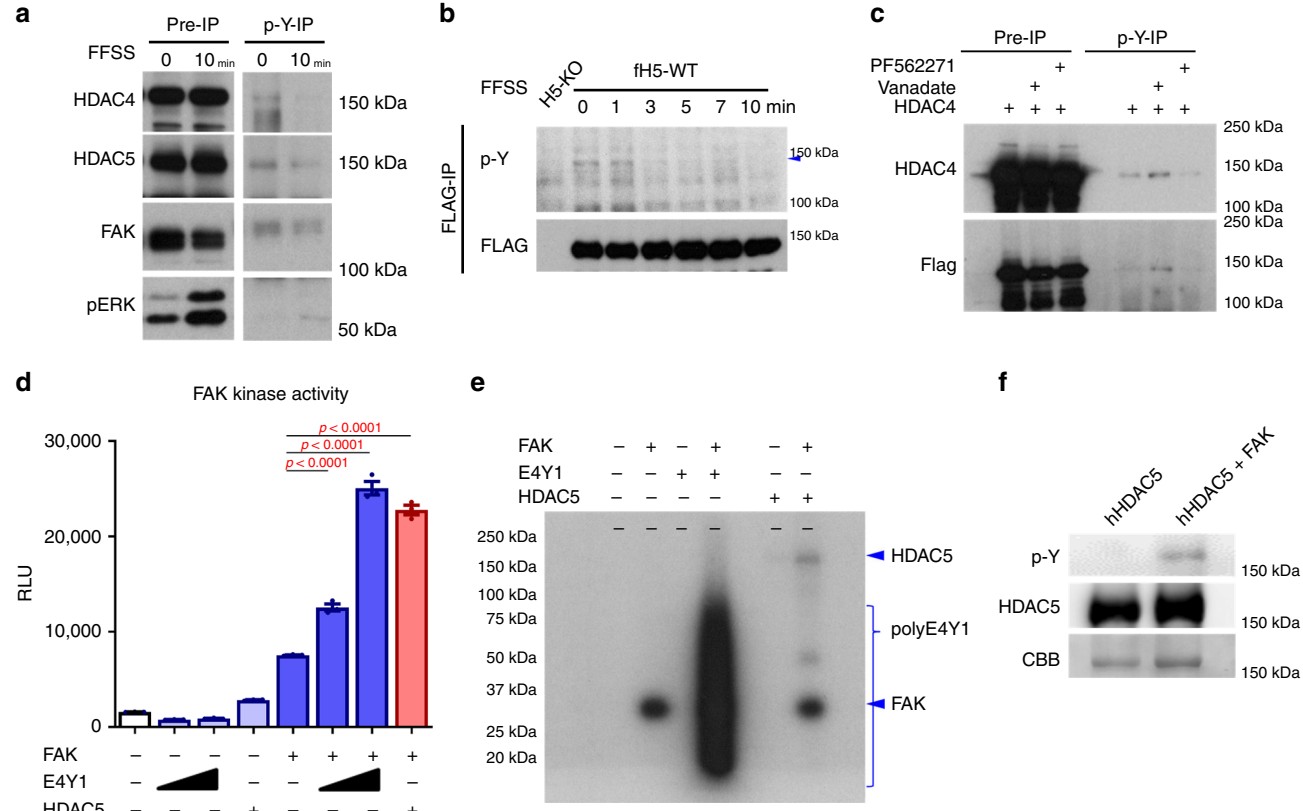

**Fig. 5 FAK-mediated class IIa HDACs tyrosine phosphorylation is regulated by FFSS. a** Immunoprecipitation by anti-phosphotyrosine (referred to as "p-Y-1000" throughout) antibody. FFSS was performed for 10 min with Ocy454 cells. Phosphotyrosine immunoprecipitation was performed on cell lysates treated as indicated, and protein expression was determined by western blotting. FFSS reduced HDAC4 and HDAC5 with p-Y-1000 immunoprecipitation. **b** FLAG immunoprecipitation was performed in HDAC5-deficient cells stably expressing FLAG-HDAC5, subjected to FFSS for times as indicated. phospho-HDAC5 was decreased in a time-dependent manner. **c** HEK293T cells were transfected with FLAG-HDAC4, then treated for one hour as indicated with PF562271 (10 μM) or vanadate (1 mM) followed by phosphotyrosine immunoprecipitation and then immunoblotting. Tyrosine-phosphorylated HDAC4 was increased by vanadate treatment and decreased by FAK inhibitor treatment. *P*-values adjusted for multiple comparisons vs control are shown. **d** FAK kinase activity was measured by ADP-Glo kinase assay. E4Y1 (poly(Glu)/poly(Tyr) ratio of 4:1) polypeptides were used as a control substrate of FAK tyrosine kinase. Increased FAK kinase activity was detected when recombinant human HDAC5 protein was used as a substrate. **e** Recombinant human HDAC5 and E4Y1 (a control substrate) were incubated with γ-32P-ATP and FAK tyrosine kinase for 1 h, then separate by SDS-PAGE followed by autoradiography. FAK treatment showed γ-32P-ATP-positive band at the expected HDAC5 recombinant protein size. **f** Kinase assay reactions as in **e** were separated by SDS-PAGE followed by immunoblotting as indicated. CBB indicates Coomassie Brilliant Blue stain. One-way ANOVA followed by Tukey–Kramer post hoc test was used (**d**). Data are expressed as mean ± SEM. Each experiment was repeated three times (**a–c, e, f**). Source data are provided as a Source Data file.

phosphorylation in osteocytes, we re-introduced HDAC5^Y642F into HDAC5-deficient Ocy454 cells via lentiviral transduction (Fig. 6e). Compared with WT HDAC5, HDAC5^Y642F shows increased nuclear localization as assessed by subcellular fractionation and immunofluorescence (Fig. 6f, g). Consistent with this, HDAC5^Y642F suppresses SOST expression more potently than WT HDAC5 when the two variants are re-introduced into HDAC5-deficient cells (Fig. 6h). Taken together, these results indicate that class IIa HDACs are tyrosine-phosphorylated in a FAK-dependent, FFSS-sensitive manner. FAK-mediated HDAC4/5 tyrosine phosphorylation is likely to play a key role in determining class IIa HDAC subcellular localization, as evidenced by dramatic alterations in HDAC4/5 immunocytochemistry staining observed in FAK-deficient cells (Supplementary Fig. 6C).

**Constitutive outside-in integrin signaling regulates the FAK/class IIa HDAC/Sost axis in osteocytes.** In many systems, FAK activity is constitutively low, then activated by increased cell-matrix interactions[43]. Osteocytes are somewhat unique in that

their dendritic projections are constitutively tethered to bone matrix[4]. In fact, one popular model of osteocyte mechanotransduction is that loading-induced changes in FFSS across osteocyte dendrites leads to transient disruptions in cell/matrix association[25]. Like many osteoblastic/osteocytic cell lines, Ocy454 cells produce high levels of the extracellular matrix[23,47]. Therefore, we hypothesized that constitutive outside-in ECM/integrin signaling leads to basal FAK activity in Ocy454 cells, which may be disrupted by FFSS. To test this model, cells were treated with the RGD peptide cilengitide[61] to disrupt integrin/ECM interactions. As predicted, cilengtide regulates expression of FFSS-responsive genes (including *Sost*) in a manner similar to (though less potent than) FAK inhibitors (Fig. 7a). Cilengtide-mediated *Sost* regulation in Ocy454 and Saos2 cells is accompanied by time- and dose-dependent reductions in FAK activity, as measured by FAK Y397 autophosphorylation, Paxillin Y118 phosphorylation, and HDAC5 Y642 phosphorylation (Fig. 7b, d). Here, human Saos2 cells were also used due to species-dependent effects of RGD peptides[62]. Cilengtide-mediated *Sost* suppression is HDAC4/5-dependent

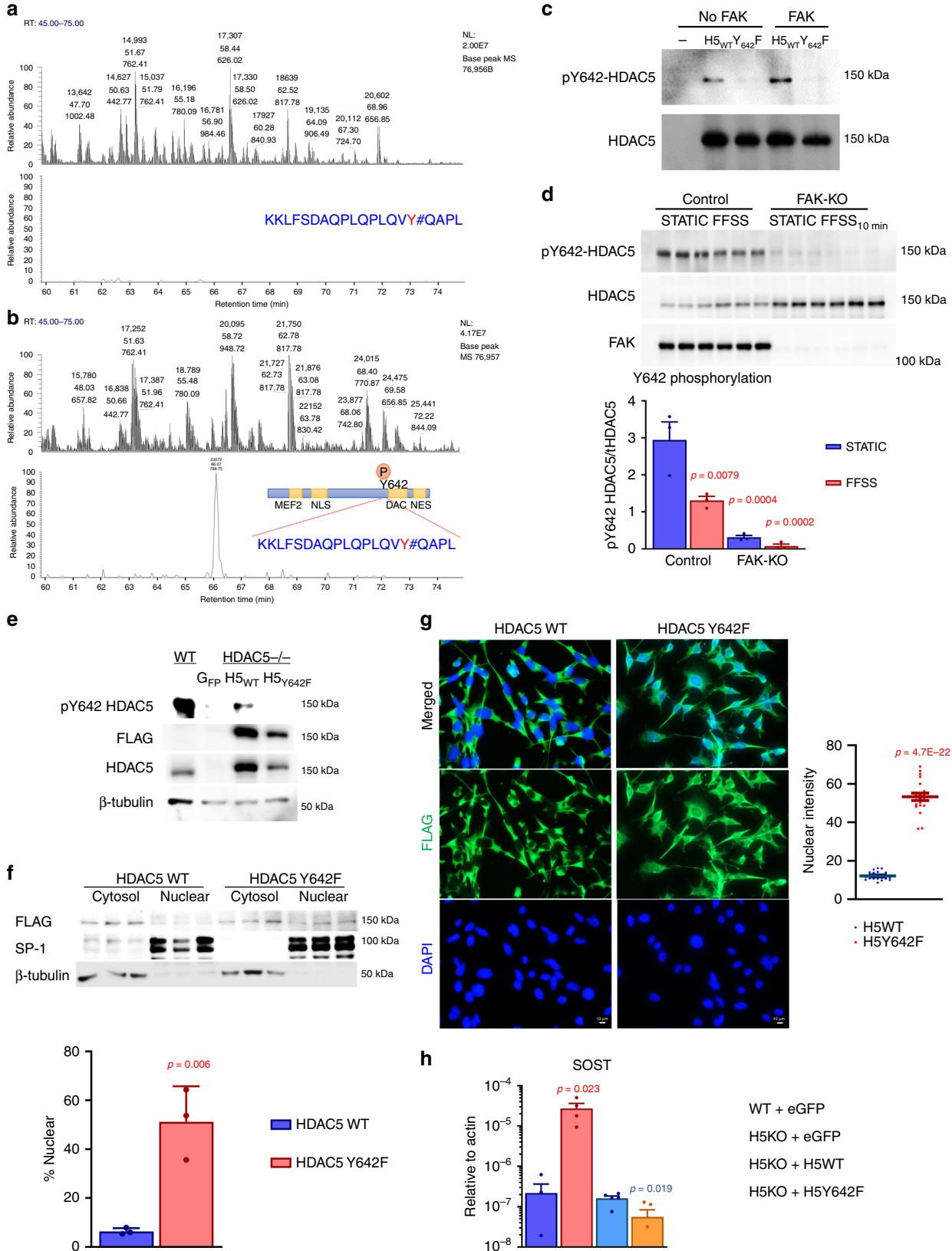

(Fig. 7e), similar to what we observed with mechanical loading (Fig. 1d), FFSS (Fig. 2a), and FAK inhibitors (Fig. 4c).

Ocy454 cells express multiple integrin isoforms predicted to bind collagen-rich ECM (Supplementary Fig. 8A, B). Deletion of single integrin isoforms failed to fully recapitulate the low *Sost* phenotype seen in FAK-mutant cells (Supplementary Fig. 8C), indicating potential redundancy amongst integrin isoforms responsible for the effects of FFSS and cilengitide.

**Fig. 6 FAK-dependent HDAC5 Y642 phosphorylation controls HDAC5 function and SOST expression. a, b** Recombinant HDAC5 was phosphorylated in vitro by FAK followed by phospho-modification analysis by mass spectrometry. Tyrosine 642 on HDAC5 was phosphorylated in a FAK-dependent manner. Top panel: the total base peak chromatogram of the chymotrypsin digested HDAC5 in the absence of FAK treatment (**a**) or presence of FAK treatment (**b**). Bottom panel: extracted ion chromatogram for the m/z value (788.75) of the phosphorylated peptide KKLFSDAQPLQPLQVY#QAPL (# symbol represents phosphorylation) demonstrating the ability to detect the phosphorylated peptide upon FAK treatment (**b**). **c** 293T cells were transfected with FLAG-tagged WT and Y642F HDAC5 cDNAs followed by anti-FLAG immunoprecipitation and then elution with FLAG peptide. Eluted protein was then used as a substrate for in vitro kinase assays plus/minus recombinant FAK followed by immunoblotting as indicated. HDAC5 WT, but not Y642F mutant, is recognized by HDAC5 pY642 antibody. Each experiment was repeated three times. **d** Control and single-cell FAK-knockout cells were treated plus/minus FFSS (10 min) followed by immunoblotting as indicated. HDAC5 Y642 phosphorylation is reduced by FFSS, and dramatically reduced in FAK-mutant cells, as quantified in the bottom panel. (*n* = 3 biologic replicates for protein were performed) *P*-values adjusted for multiple comparisons vs control in STATIC conditions are shown. **e** HDAC5-deficient Ocy454 cells were reconstituted with FLAG-tagged lentiviral constructs followed by immunoblotting as indicated. **f** Lentiviral reconstituted HDAC5-deficient cells were subjected to subcellular fractionation followed by immunoblotting as indicated. The percentage of the total FLAG-HDAC5 (WT or Y642F) in the nuclear fraction was measured by densitometry. (*n* = 3 biologic replicates for protein were performed) *P*-values vs HDAC5 WT. **g** Cells as in **f** were subjected to anti-FLAG immunocytochemistry. Nuclear FLAG intensity was quantified (*n* = 20 cells were analyzed). HDAC5$^{Y642F}$ shows increased nuclear localization compared to HDAC5$^{WT}$ grown under identical conditions. **h** Cells as in **f** were grown at 37 °C for 14 days. RNA was isolated for RT-qPCR. HDAC5-deficient cells show increased SOST expression which is reduced to a greater extent by HDAC5$^{Y642F}$ than HDAC5$^{WT}$ reconstitution. (*n* = 4 biologic replicates for RNA for WT + eGFP and H5KO + eGFP) *P*-values (red) vs WT + eGFP. (*n* = 5 biologic replicates for RNA for H5KO + H5WT and H5KO + H5Y642F) *P*-values (blue) vs H5KO + H5WT. Two-sided unpaired *t* test (**f, h**) and one-way ANOVA followed by Tukey–Kramer post hoc test (**d**) were used. Data are expressed as mean ± SEM. Source data are provided as a Source Data file.

**FAK inhibitors mimic the effects of mechanical loading in vivo.** Finally, we sought to test the effects of small molecule FAK inhibitors on osteocyte gene expression in vivo. Mice were treated with a single intraperitoneal injection of VS-6063 (60 mg/kg) and sacrificed 5 h later. VS-6063 activity was first confirmed by demonstrating reduced FAK Y397 autophosphorylation in liver extracts (Fig. 8a, each lane shows liver extracts from a different mouse) and bone tissue (Fig. 8c). RT-qPCR was then performed on cortical bone for *Sost*, and additional genes previously demonstrated to be regulated by FFSS and FAK inhibitors in Ocy454 cells. Consistent with our in vitro cell culture data, in vivo VS-6063 treatment significantly regulates expression of *Sost*, *Ankrd37*, *Apln*, *Mss51*, and *Hoxb5* (Fig. 8b), all in the predicted direction. Consistent with reduced *Sost* mRNA, VS-6063 treatment reduced sclerostin levels by osteocytes in situ (Fig. 8d).

Taken together, these results support a model (Fig. 8e) in which constitutive outside-in matrix/integrin signaling leads to basal FAK activation in osteocytes. FAK can directly phosphorylate class IIa HDACs and regulate their subcellular localization. FFSS disrupts cell/matrix interactions and therefore transiently blocks FAK-mediated class IIa HDAC tyrosine phosphorylation. Class IIa HDACs play a crucial role in FFSS-mediated *Sost* suppression. Our data suggest multiple potential interventions (RGD peptides and FAK inhibitors) to simulate or augment osteocyte mechanotransduction.

## Discussion

Sensing changes in diverse mechanical cues is crucial in physiology and pathophysiology[63]. Of all mechanosensitive cell types, osteocytes are relatively unique in that they are deeply buried within ECM-rich mineralized bone tissue. Therefore, osteocytes are well-suited to adapt to dynamic, mechanosensitive changes in cell/matrix interactions. Here, we observe that high basal FAK activity in osteocytes is blocked by disrupting cell/matrix interactions with RGD peptides, and that cilengitide treatment triggers HDAC4/5-mediated *Sost* suppression. Moreover, FAK directly phosphorylates HDAC4 and HDAC5, indicating that tyrosine phosphorylation of class IIa HDACs represents a mechanism controlling the subcellular localization of these key MEF2-regulating factors.

Previous studies have demonstrated that dynamic changes in serine/threonine class IIa HDAC phosphorylation are linked to changes in 14-3-3 binding and subcellular localization. Upstream signaling pathways including cAMP/protein kinase A/salt-inducible kinases[24,64] and calcium/calmodulin-dependent kinases[65] lead to HDAC4/5 serine/threonine phosphorylation and cytoplasmic retention. Here, we demonstrate that FAK-mediated tyrosine phosphorylation also controls HDAC4/5 subcellular localization. Class IIa HDACs are well-poised to integrate gene expression changes in response to diverse hormonal and mechanical cues. Interestingly, previous studies have suggested that FFSS signaling in other cell types (endothelial cells and renal epithelial cells) trigger class IIa HDAC movement from the nucleus to the cytoplasm[66–68]. Therefore, it is likely that the distinct mechanisms through which mechanical cues control class IIa HDAC subcellular localization are likely to be cell type-specific, and controlled by the nature of cell/matrix interactions in a particular cell and tissue type. This observation is internally consistent with our in vivo observations. At the strain magnitude induced in this study, the cantilever bending model is characterized by the induction of periosteal osteoblast function[44]. Though many aspects of mechanical loading (including direct tissue strain and secondary interstitial fluid flow) are known to be anabolic to bone cells, it is not possible to decouple these stimuli in vivo. However, in this context, the heterogeneous fluid-flow stimuli induced via external skeletal loading[69–71] is one stimulus known to activate both osteocytes and differentially activate bone surface cells[72]. The precise link between the osteocyte mechanotransduction signaling cascade described here and skeletal responses to loading in vivo remains to be unraveled.

Here, we identified Y642 as a FAK-dependent HDAC5 tyrosine phosphorylation site in vitro and in cultured osteocytes. FAK-mutant cells show increased HDAC5 nuclear localization. However, the precise role of HDAC5 Y642 phosphorylation in controlling its subcellular localization remains to be determined. We note that Y642 is in close proximity to HDAC5 S661, a site that when phosphorylated leads to 14-3-3 binding and cytoplasmic HDAC5 retention[73]. Future studies are needed to define the exact role of HDAC5 tyrosine phosphorylation at this site, and to determine the relationship between Y642 phosphorylation and nearby serine 661 phosphorylation and 14-3-3 binding. Furthermore, future studies are required to understand the mechanism through which FAK deficiency results in increased HDAC5 mRNA (Supplementary Dataset 1) and protein (Fig. 6d) levels.

Of the four class IIa HDAC proteins, only HDAC5 is linked to bone mass in mice[23] and humans[74]. HDAC9 expression is

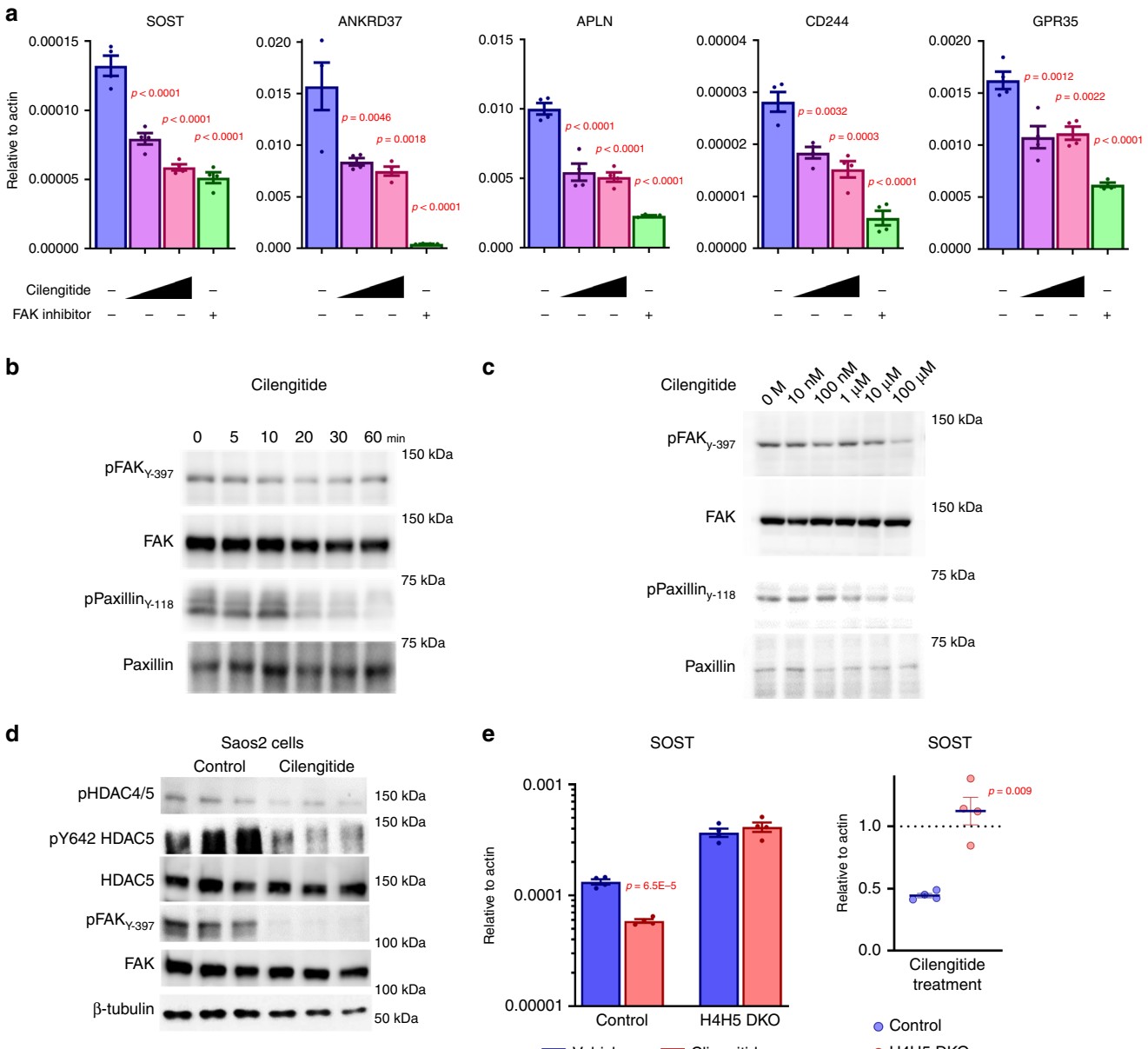

**Fig. 7 RGD peptide blocks FAK activity and reduce Sost in a HDAC4/5-dependent manner. a** Ocy454 cells were treated with cilengitide (10 µM and 50 µM) or PF562271 (10 µM) for 4 h, followed by RT-qPCR for FFSS-regulated genes as indicated. Both cilengitide and PF562271 regulate expression of FFSS-responsive genes. ($n = 4$ biologic replicates for RNA) P-values adjusted for multiple comparisons vs controls are shown. **b** Cells were treated with cilengitide (50 µM) for the indicated times followed by immunoblotting. **c** Cells were treated with the indicated doses of cilengitide (1 h) followed by immunoblotting. pHDAC4/5 immunoblotting was performed using an antibody that recognizes HDAC4 pS246 and HDAC5 pS259. **d** Saos2 cells were treated with cilengitide (100 µM) for 60 min followed by immunoblotting. **e** WT and HDAC4/5 double-knockout (H4H5-DKO) cells were treated with vehicle or cilengitide (50 µM) for 4 h followed by *Sost* RT-qPCR. Cilengitide treatment decreased *Sost* expression in control cells, but not in H4H5-DKO cells. ($n = 4$ biologic replicates for RNA) P-values vs control in the left and right panels. Two-sided unpaired *t* test **e** and one-way ANOVA followed by Tukey–Kramer post hoc test (**a**) were used. Data are expressed as mean ± SEM. Each experiment was repeated three times (**b-d**). Source data are provided as a Source Data file.

exceptionally low in Ocy454 cells (Supplementary Dataset 1), and compound deletion of both HDAC4 and HDAC5 is sufficient to block loading-induced bone formation (Fig. 1a) and FFSS-induced sclerostin suppression (Fig. 2A). While we cannot exclude a function for HDAC7 in osteocyte mechanotransduction, our current data support a model for a major functional role of HDAC5 (with potential compensation from HDAC4) in FFSS-induced sclerostin suppression. Our compound HDAC4/5 mutant mice lack HDAC5 globally and HDAC4 in DMP1-Cre-expressing cells (mainly mature osteoblasts and osteocytes).

Therefore, we acknowledge the possibility that extra-skeletal HDAC5 expression may contribute to the phenotypes observed in vivo in Fig. 1. However, our Ocy454 cell data strongly suggests an osteocyte-intrinsic role for this protein.

While it is possible that type I collagen fibers (which are highly abundant in bone ECM) are a ligand that promotes outside-in integrin/FAK signaling in osteocytes, non-collagenous bone matrix proteins such as fibronectin and osteopontin are also highly expressed by Ocy454 cells[24]. Cilengitide can inhibit both αvβ3 and α5β1-containing integrin heterodimers (which both

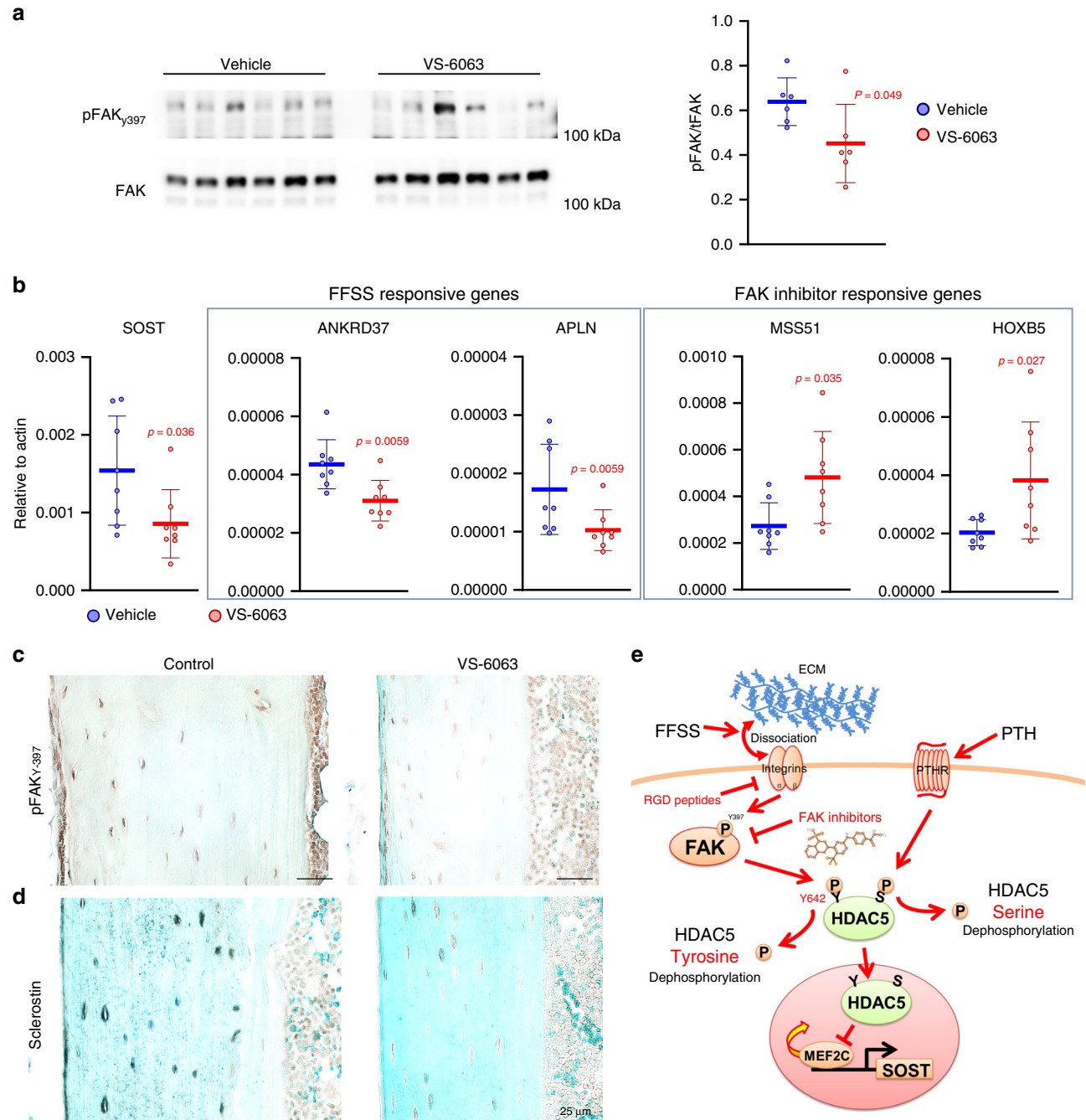

**Fig. 8 FAK inhibitors reduce osteocytic Sost expression in vivo. a** FAK inhibitor (VS-6063) was administrated by single intraperitoneal injection (60 mg kg$^{-1}$), and liver protein lysates were collected 5 h later and then subjected to immunoblotting as indicated. *P*-values vs vehicle. *N* = 6. **b** Cortical bone RNA was isolated after acute VS-6063 treated followed by RT-qPCR as indicated. FAK inhibitor significantly regulated expression of FFSS- and FAK inhibitor-sensitive transcripts. FFSS-responsive genes are those that were regulated by FFSS in vitro in our RNA-seq results. FAK inhibitor responsive genes are those that were upregulated by PF562271 in vitro in our RNA-seq results. *P*-values vs vehicle, *n* = 8 mice. **c, d** Immunohistochemistry was performed on tibia samples from mice in (**a**) and (**b**). Immunohistochemistry of mouse long bone. **c** phospho-FAK (Y397) and **d** Sclerostin levels were downregulated by FAK inhibitor treatment. **e** Model demonstrating the effects of fluid-flow shear stress (FFSS)-mediated gene regulation in osteocytes, see text for details. Data are expressed as mean ± SEM. Each experiment was repeated three times (**c, d**). Source data are provided as a Source Data file.

bind fibronectin and osteopontin) with nanomolar IC$_{50}$ values[61], supporting the use of this RGD peptide in our studies. The precise identity of the ECM component/receptor pair that drives basal FAK signaling in osteocytes remains to be determined. Moreover, other relevant non-fibrillar FAK-activating integrin ligands may exist. One interesting example is Irisin, a myokine whose secretion is increased in response to exercise[75]. Recently, Irisin-mediated FAK activation in osteocytes was suggested downstream of binding to an αV/ß5 integrin receptor[76]. Irisin-triggered FAK activation in osteocytes was reported to increased *Sost* levels, findings consistent with our model here that FAK inhibition reduces *Sost* expression. Therefore, our current work provides a conceptual framework to explore future studies investigating the relationship between FFSS and Irisin signaling in osteocytes.

As detailed above, multiple upstream signaling pathways are engaged as osteocytes respond to external cues[4]. Since *Sost* gene regulation is central to the skeletal response to loading in vivo, we have focused on characterizing the signaling steps that are necessary for FFSS-induced *Sost* suppression in vitro and in vivo. That being said, FAK- and class IIa HDAC-independent osteocyte mechanotransduction mechanisms are likely to exist, just as *Sost*-independent skeletal adaptation has been reported[77]. In addition to integrating our current findings with previously-described membrane proximal changes in response to FFSS in osteocytes, future studies will be needed to better understand how local dynamic changes in FAK activity in osteocyte dendrites ultimately reach the nucleus to regulate gene expression. Recently, the mechanosensitive cation channel Piezo1 has been shown to play a key role in skeletal homeostasis[78–81]. However, direct links between mechano-dependent Piezo1 signaling and the FFSS/FAK/HDAC5/SOST axis described here remain to be determined.

FAK is a broadly expressed cytosolic tyrosine kinase. Previous studies identified a mechanotransduction-independent role of FAK in regulating osteocyte cell survival[82]. Moreover, FAK-deficient osteoblasts fail to induce immediate early genes in response to fluid flow in vitro[83]. FAK conditional knockout mice using Col1a1-Cre were reported to show no skeletal phenotype at baseline or in response to external loading[84]. However, these studies did not assess deletion efficiency with Col1a1-Cre (which is known to be inefficient for certain "floxed" alleles) or compensation by the closely related kinase Pyk2. In addition, Pyk2 is highly expressed in osteoclasts and plays a key role in osteoclast cytoskeletal reorganization during bone resorption[85]. A FAK/PYK2 inhibitor (PF431396) showed increases bone formation and bone mass in female rats subjected to surgical removal of the ovaries[86]. In this study, the mechanism by which PF431396 increases bone formation was not fully understood. Our current work (see Fig. 8) indicates that lowering sclerostin levels may be a part of the osteoanabolic mechanism associated with systemic FAK inhibitor treatment. FAK inhibitors are currently under investigation for cancer[87] and tissue fibrosis[88]. FAK inhibitors quickly reduce phosphorylation levels of FAK substrates, suggesting that high constitutive cellular phosphatase activity is present. Indeed, rapid "on/off" phosphorylation–dephosphorylation cycles are thought to contribute to rapid and robust cellular signaling output (reviewed in REF. [89]). Based on their ability to mimic effects of FFSS in osteocytes, bone-targeted FAK inhibitors may represent anabolic osteoporosis treatment agents.

Taken together, our results demonstrate that a FAK/class IIa HDAC axis plays a crucial role in osteocyte mechanotransduction in vitro and in vivo. These findings highlight a central role for class IIa HDACs as signaling molecules that integrate inputs from a wide variety of external cues into gene expression changes. Moreover, this work opens up avenues toward modulating osteocytic sclerostin production to boost bone mass for diseases, including osteoporosis.

## Methods

**Genetically modified mice**. All animals were housed in the Center for Comparative Medicine at the Massachusetts General Hospital ($21.9 \pm 0.8$ °C, $45 \pm 15\%$ humidity, and 12-h light cycle 7 am–7 pm), and all experiments were approved by the hospital's Subcommittee on Research Animal Care. The following published genetically modified strains were used: *Hdac4* floxed mice (RRID: MGI:4418117)[90], germline *Hdac5*-deficient mice (RRID: MGI:3056065)[91], and DMP1-Cre mice (RRID: MGI:3784520)[92]. Mice were backcrossed to C57BL/6 mice for at least seven generations. Floxed littermates without Cre were used as wild-type controls. Genotypes were determined by PCR using primers described in SupplementaRY Dataset 2. Power calculations were performed based on pilot experiments.

**Chemicals**. Chemical reagents U0126 (Cell Signaling Technology, #9903), PD98059 (Cell Signaling Technology, #9900), Y-27632 (Cell Signaling Technology, #13624), SB431542 (Selleckchem, S1067), CCG1423 (Selleckchem, S7719),

verteporfin (Sigma, SML0534). VS-6064 (Selleckchem, S7654), PF431396 (Selleckchem, S7644), PF562271 (Selleckchem, S2890), and Vanadate (New England Biolabs, P0758) were used at doses and times indicated in figure legends. All compounds were dissolved in DMSO.

**In vivo FAK inhibitor treatment**. To assess the acute effect by FAK inhibitors (VS-6063), C57BL/6 mice at 6–7-weeks old were treated with a single intraperitoneal administration at 60 mg kg$^{-1}$ (dissolved in sunflower oil).

**In vivo loading studies**. For the acute and long-term mechanical loading studies, wild-type, HDAC4 conditional knockout (cKO, HDAC4$^{f/f}$, DMP1-Cre), HDAC5 knockout, and double-knockout (DKO, HDAC4$^{f/f}$; HDAC5$^{-/-}$; DMP1-Cre) female mice (20 wk) underwent cyclical loading of their right tibia using a non-invasive murine tibial cantilever bending device (SupplementaRY Fig. 1[93]). With the mouse anesthetized with inhaled isoflurane (2%), the proximal tibia is secured via a gripping bracket. A loading tine was then used to apply force to the distal tibia (medial/lateral plane) via a computer controlled linear force actuator. As described previously[44], we used beam theory to estimate the end load required to induce a given magnitude of peak normal strain at the mid-shaft of each mouse (2500 µε). Mice were assigned to either acute (three loading bouts on consecutive days) or long-term (nine loading interventions, three bouts/week) protocols, and, in both studies, were sacrificed 24 h after the final round of loading. Each loading bout consisted of 100 cycles applied at 1 Hz. For the acute protocol, cortical bone RNA was isolated and *Sost* expression determined by RT-qPCR. Mice assigned to the 3-week protocol received Calcein labels on d10 and d19. Periosteal osteoblastic activity was determined by characterizing periosteal mineralizing surface (p.MS/BS), mineral apposition rates (p.MAR) and periosteal bone formation rate (p.BFR/BS) at the tibia mid-shaft of contralateral (left) and experimentally loaded (right) tibia using standard dynamic histomorphometry procedures via a custom ImageJ software[94,95].

**Dynamic skeletal histomorphometry**. All mice received intraperitoneal calcein injections (15 mg/kg) 11 days and 2 days prior to sacrifice. As in previous studies[44], right and left tibiae were dissected, and 300-µm-thick sections spanning the tibia mid-shaft were obtained via a Struers minitome. Sections were hand ground to 100 µm. Digital images were obtained via fluorescent microscopy, and were blinded and dynamic histomorphometry outcomes assessed for both endocortical and periosteal surfaces. Specifically, we determined s.LS (peripheral Single-Labeling surface), d.LS (peripheral double-labeling surface), and interlabel thickness (Ir.L.Th) for each surfaces. From these data, mineralizing surface (MS; MS = (dLS + sLS/)/bone surface [BS] × 100), MAR (mineral apposition rate (MAR; MAR = Ir.L.Th/interlabel time period [Ir.L.t]), and bone-mineral formation rate (BFR; BFR = MAR × MS/BS) were calculated per ASBMR guidelines[96].

**Femur micro-CT and mechanical testing**. Micro-CT imaging was performed on a bench-top scanner (µCT 40, Scanco Medical AG, Brutisellen, Switzerland) to measure the morphology of the femoral mid-diaphysis prior to mechanical testing (Scanning parameters: 10-µm isotropic voxel, 70 kVp, 114 mA, 200-ms integration time). A 500-µm long region of the mid-diaphysis was scanned, bone segmented from surrounding tissue using a threshold of 700 mgHA/cm$^3$ and the geometry of the cortex analyzed using the Scanco Evaluation Program. Femora were mechanically tested in three-point bending using a materials testing machine (Electroforce 3230, Bose Corporation, Eden Prairie, MN). The bending fixture had a bottom span length of 8 mm. The test was performed in displacement control moving at a rate of 0.03 mm/s with force and displacement data collected at 50 Hz. All bones were positioned in the same orientation during testing with the cranial surface resting on the supports and being loaded in tension. Bending rigidity (EI, N-mm$^2$), apparent modulus of elasticity ($E_{app}$, MPa), and ultimate moment ($M_{ult}$, N-mm) were calculated based on the force and displacement data from the tests and the mid-diaphysis geometry measured with µCT. Bending rigidity was calculated using the linear portion of the force-displacement curve. The minimum moment of inertia ($I_{min}$) was used when calculating the apparent modulus of elasticity.

**Cell culture**. A single-cell sub-clone of Ocy454 cells[23] was used for all experiments. Saos2 cells were from ATCC. Cells were passaged in alpha-MEM supplemented with heat inactivated 10% fetal bovine serum and 1% antibiotic–antimycotic (Gibco$^{TM}$) at 33 °C with 5% $CO_2$. Cells were seeded at 50,000 cells/mL and allowed to reach confluency at 33 °C in 2–3 days. Then, cells were transferred from 33 °C to 37 °C to inactivate the temperature-sensitive T antigen and facilitate osteocytic differentiation. For protein and gene expression analyses, cells were analyzed after culture at 37 °C for 14–21 days. Cells were routinely assessed by *Sost* and *Dmp1* expression at 37 °C and examined for osteocytic morphology. For FFSS experiments, cells were changed to medium containing HEPES (25 mM) the day prior to experiments. Cells were placed on a Thermomixer platform (Eppendorf, ThermoMixer R) heated to 37 °C and subjected to orbital rotation (350 rpm, ~6.56 dynes/cm$^2$) followed by RNA or protein isolation. For two-dimensional laminar fluid shear stress, Ocy454 cells were plated on glass microscope culture slides (Flexcell International Corp.) at $2 \times 10^5$ cells/ml and allowed to grow at the

permissive temperature (33 °C) for 3 days. Subsequently, slides were moved to 37 °C for an additional culturing time (11–14 days). Medium was changed for static slides, or slides were loaded into the laminar fluid-flow shear stress device (Flexcell Streamer, Flexcell International Corp.) connected to an electronically controlled peristaltic pump with pulse dampers integrated into the flow circuit to allow for continuous unidirectional shear stress of 8 dynes/cm$^2$ for 45 min.

For experiments in 293T cells, 1 μg of FLAG-hHDAC4 plasmid (Addgene 30485) was transfected using Fugene HD (Promega). Forty-eight hours later, protein lysate was collected with 1 mL of TNT (Tris-NaCl-Tween buffer, 20 mM Tris-HCl pH 8, 200 mM NaCl, 0.5% Triton X-100 containing protease inhibitors (PI), 1 mM NaF, 1 mM DTT, 1 mM vanadate). Fifty μL of protein lysate was saved as "pre-IP" sample. The remaining protein lysate was incubated with 1 μg of p-Y-1000 antibody (Cell Signaling Technology, #8954) overnight after which 50 μL protein A/G agarose beads (Pierce, #20421) were added and rotated for 2 h. TNT buffer was washed two times containing PI, NaF, DTT, and vanadate. After spinning down beads, sample was suspended 50 μL SDS sample buffer and heated at 95 °C for 5 min. For FLAG IP experiments, HDAC5-KO Ocy454 cells (see lentiviral transduction and CRISPR-cas9-mediated gene deletion) were infected with lenti-GFP or lenti-FLAG-human HDAC5 vectors. Cells were analyzed after culture at 37 °C for 14–21 days, and were subjected to FFSS. Protein lysate was collected with TNT buffer (containing PI, NaF, DTT, and vanadate), and incubated with anti-FLAG affinity gel (Bimake, Houston, TX, B23101) overnight to purify FLAG-hHDAC5 proteins for p-Y-1000 immunoblotting.

**Lentiviral transduction and CRISPR/Cas9-mediated gene deletion.** For sgRNA experiments, Ocy454 cells were stably transduced with a blasticidin-resistant Cas9 expressing lentivirus which caused no discernable effects on sclerostin production, PTH responsiveness, or FFSS-responsiveness. sgRNA sequences were subcloned into pLentiGuide-Puro (Addgene; plasmid# 52963) plasmid (see Supplementary Dataset 2 for sgRNA sequences used). To design sgRNA target sequences, we used "Design sgRNA for CRISPRko" web tool (http://portals.broadinstitute.org/gpp/public/analysis-tools/sgrna-design) and selected top two guide sequences for cloning. To produce lentiviruses, Ocy454 cells transfected with pLentiGuide, psPAX2 (Addgene; plasmid #12260), and MD2.G (Addgene; plasmid #12259) using Fugene HD (Promega, WI, US). Medium was changed the next day, and then collected 48 h later. Cells were exposed to lentiviral particles overnight at 33 °C in the presence of polybrene (2.5 μg mL$^{-1}$). Media was then changed with puromycin (4 μg mL$^{-1}$) and blasticidin (5 μg mL$^{-1}$). Cells were maintained in selection medium throughout the duration of the experiment. Some bulk cell populations were seeded in 96-well plates at 0.2 cell/well. Media were changed twice a week, and 3 weeks later single-cell colonies were identified by microscopy. Single-cell-derived colonies were expanded and analyzed for loss of target protein expression by immunoblotting. For FAK CRISPR-KO experiments, at least six independent clones from each bulk sgRNA cell line were tested by immunoblotting, and clones with 100% protein deletion were used for experiments. Consistent results with respect to reduced SOST expression were seen in all single-cell clones where FAK protein was absent by immunoblotting. FAK-KO clone 1–4 (FAK-KO c1–4), derived from the bulk population of cells transduced with sgRNA-1, was used for subsequent experiments. Control cells were transduced with an empty sgRNA-expressing lentivirus.

HDAC5-deficient cells as described[23] were infected with FLAG-tagged HDAC5 variants in a pLX_311 backbone (Addgene plasmid 118018) followed by blasticidin selection. HDAC5 S259/498A mutant cDNA was obtained from Addgene (plasmid 32216), HDAC5 Y642F construct was synthesized de novo (VectorBuilder).

**RNA isolation and transcript analyses.** The total RNA was collected from cultured cells using QIAshredder (QIAGEN) and PureLink RNA mini kit (Invitrogen$^{TM}$) following the manufacturer's instructions. Quickly, lysis buffer with 2-mercaptoethanol was added to cold PBS-washed cells and collected into QIAshredder, then centrifuged at 15,000 g for 3 min. The flow-through was collected into a new tube, and RNA isolation was carried out with PureLink RNA mini kit following the manufacturer's instructions. For long bone isolation, mice were sacrificed, and both tibias (for mechanical loading tests) and both femurs (for FAK inhibitor treatment) were dissected out. Soft tissue was removed, and epiphysis cut. Bone marrow cells were removed by centrifugation and flushing with cold DPBS. The bone samples were quickly snap frozen in liquid nitrogen and stored at −80 °C. RNA was extracted by tissue blender with TRIzol (Life technologies) following the manufacture's instruction, and further purification was performed with PureLink RNA mini column. For qRT-PCR, cDNA was prepared with 750 ng of RNA using the Primescript RT kit (Takara Inc.) and analyzed with PerfeCa® SYBR® Green FAstMix® ROX (Quanta bio) in the StepOnePlus$^{TM}$ Real-time PCR System (Applied Biosystems) using specific primers designed for each targeted gene. Relative expression was calculated using the $2^{-\Delta\Delta CT}$ method by normalizing with β-actin housekeeping gene expression, and presented as fold increase relative to β-actin.

**Cell viability assays.** Ocy454 cells were cultured in 96-well plates at 37 °C for 7 days, and then were treated with different doses of FAK inhibitors (PF562271, VS-6063 and PF431396) for 4 h. After treatment, media was changed, and cells

were incubated for 1 h at 37 °C with fresh 10% PrestoBlue solution (PrestoBlue$^{TM}$ Viability Reagent, Life technologies) containing culture media (alpha-MEM supplemented with heat inactivated 10% fetal bovine serum and 1% antibiotic–antimycotic, Gibco$^{TM}$). Absorbance was read at 570 nm (resavurin-based color change) and 600 nm (background) to calculate cell viability per the instructions of the manufacturer.

**RNA-seq analysis and bioinformatics.** RNA sequencing was conducted using a BGISEQ500 platform (BGI, Shenzhen, China)[97]. Briefly, RNA samples with RIN values >8.0 were used for downstream library construction. mRNAs were isolated by PAGE, followed by adaptor ligation and RT with PCR amplification. PCR products were again purified by PAGE, and dissolved in EB solution. Double-stranded PCR products were heat denatured and circularized by the splint oligo sequence. The ssCir DNA was formatted as the final sequencing library, and validated on bioanalyzer (Agilent 2100) prior to sequencing. The library was amplified with phi29 to general DNA nanoballs (DNBs), which were loaded into the patterned nanoarray followed by SE50 sequencing. On average, we obtained 20 M reads per bone RNA sample. Sequencing reads were mapped by the STAR aligner[98] to the mm9 reference genome using Ensembl annotation. Gene expression counts (CPM or RPKM) were calculated using HTSeq v.0.6.0[99]. Genes with expression counts of zero for two or more samples were removed. Differential expression analysis was performed using EgdeR package[100] based on the criteria of more than twofold change in expression value versus control and false discovery rates (FDR) < 0.05. For the heatmap shown in Fig. 3g, log2[CPM] values are used to calculate mean and variance along the gene axes across all samples. The heatmap.2 function from the gplots package was used to generate the heatmap in terms of Z scores. The expression across gene axis (rows) was scaled with the scaledata function using the color code indicated in the figure legends. Genes and samples were reordered by hierarchical clustering. The RNA-seq data have been deposited to GEO under accession numbers GSE139604 and GSE144265. Supplementary Dataset 1 shows RPKM, log2FC, and FDR values for all RNA-seq studies performed. Supplementary Dataset 3 shows quality control metrics for all libraries that were sequenced.

Analysis of gene ontology enrichment among 369 differentially expressed genes (DEGs) was performed using Enrichr (https://amp.pharm.mssm.edu/Enrichr/) on the GO Cellular Component 2017b set. The combined EnrichR score based on Z scores and P-values was used in graphic representation. In addition, the pathway enrichment analysis among DEGs was performed using the GeneXplain 4.8 and Ingenuity pathway analysis (IPA) tools, with statistical significance of enrichment visualized as −log10(P-value). Statistical significance of the overlap between lists of DEGs was assessed by hypergeometric test based on 12,849 expressed genes using an online tool (nemates.org/MA/progs/overlap_stats.html).

**Western blotting.** Whole-cell lysates were prepared using TNT (Tris-NaCl-Tween buffer, 20 mM Tris-HCl pH 8, 200 mM NaCl, 0.5% Triton X-100 containing protease inhibitor (PI), 1 mM NaF, 1 mM DTT, 1 mM vanadate). Adherent cells were washed with ice cold PBS, then scraped into TNT buffer on ice. Material was then transferred into Eppendorf tubes kept on ice, vortexed at top speed for 30 s, then centrifuged at 14,000 g for 6 min at 4 °C. For subcellular fractionation, cells were initially resuspended in hypotonic lysis buffer (20 mM HEPES, 10 mM KCl, 1 mM MgCl2, 0.1% Triton X-100, 5% glycerol supplemented with DTT, protease inhibitors, and phosphatase inhibitors) for 5 min on ice. Nuclear pellets were spun down at 5000 g for 5 min, and the supernatant was saved as the cytoplasmic lysate. Thereafter, the nuclear pellet was washed once in 1 mL of hypotonic lysis buffer. The nuclear pellet was then resuspended in hypertonic lysis buffer (20 mM HEPES, 400 mM NaCl, 1 mM EDTA, 0.1% Triton X-100, 5% glycerol supplemented with DTT, protease inhibitors, and phosphatase inhibitors), followed by vortexing twice for 30 s. Debris was spun down at 14,000 g for 5 min, and the supernatant was saved as the nuclear lysate. For immunoblotting, lysates or immunoprecipitates were separated by SDS-PAGE, and proteins were transferred to the nitrocellulose. Membranes were blocked with 5% milk in tris-buffered saline plus 0.05% Tween-20 (TBST) and incubated with primary antibody overnight at 4 °C. The next day, membranes were washed, incubated with appropriate HRP-coupled secondary antibodies, and signals detected with ECL Western Blotting Substrate (Pierce), ECL Plus Western Blotting Substrate (Pierce), or SuperSignal West Femto Maximum Sensitivity Substrate (Thermo scientific). The primary antibodies were FAK (1:1000, Cell Signaling Technology, 13009), p-FAK(Y397) (1:500, Cell Signaling Technology, 8556), phospho-Paxillin (1:250, Cell Signaling Technology, 2541), Paxillin (1:500, Cell Signaling Technology, 12065), HDAC4 (1:1000, Abcam, ab12172), phospho-HDAC4/5/7 (S246/S259/S155) (1:250, Cell Signaling Technology, 3443), DYKDDDDK tag (1:1000, Cell Signaling Technology, 2368), p-p44/42 MAPK (T202/Y204) (pERK) (1:500, Cell Signaling Technology, 9101), p44/42 MAPK (Erk1/2) (1:1000, Cell Signaling Technology, 9102), p-Y-1000 (1:1000, Cell Signaling Technology, 8954), and beta-tubulin (1:250, Cell Signaling Technology, 5346). The HDAC5 pY642 antibody (1:1000) was produced using YenZym Antibodies' P-site™ Antibody Service protocol. The antibody was raised and affinity-purified against pTyr642-HDAC5, Hu #634~648 (QPLQPLQV-pY-QAPLSL-amide) and affinity-absorbed against the non-phosphorylated peptide to remove the cross-reactive antibody population. Uncropped western blots are available in the Source Data File.

**Immunohistochemistry (IHC).** Fomalin-fixed paraffin-embedded decalcified tibia sections were obtained from long-term loading treated mice and FAK inhibitor (VS-6063, 60 mg kg$^{-1}$) treated mice with their control groups. For anti-sclerostin immunohistochemistry, antigen retrieval was performed using proteinase K (20 µg mL$^{-1}$) for 15 min at room temperature. Endogenous peroxidases were quenched, and slides were blocked in TNB buffer (Perkin Elmer), then stained with anti-sclerostin biotinylated antibody at a concentration of 1:50 overnight at 4 °C. Sections were washed with TBST 3 times, and incubated with HRP-coupled secondary antibodies, signals amplified using tyramide sinal amplification and HRP detection was performed using 3,3′-diaminobenzidine (DAB, Vector) for 5–10 min. Slides were counterstained with 0.02% fast green staining solution or hematoxylin. For anti-active ß-catenin immunohistochemistry, a similar protocol was followed except trypsin (10 min at 37 °C) was used for antigen retrieval. Quantification of sclerostin-positive osteocytes was performed by using ImageJ software on the blind-test manner. Quantification of active ß-catenin-positive periosteal cells was performed in a similar manner using ImageJ. Here, hematoxylin counter staining was used to identify periosteal cells, defined as flat cells present on periosteal surfaces. The primary antibodies were goat anti-mSOST biotinylated (1:50, R&D systems, BAF1589), rabbit anti-non-phospho (active) β-catenin (Ser33/37/Thr41) (1:50, Cell Signaling Technology, 8814), and rabbit p-FAK (Y397) (1:50, Cell Signaling Technology, 8556).

**Immunocytochemistry.** Ocy454 cells were seeded in a eight-chamber polystyrene vessel tissue culture-treated glass slides, and cultured at 37 °C for 14 days. Immediately after Fluid-Flow Shear Stress treatment, cells were fixed with cold acetone at −20 °C for 10 min. After three times washing with DPBS, cells were blocked with 1.5% bovine serum albumin (BSA) in DPBS for 30 min. Then, slides were incubated with primary antibodies in blocking buffer (1.5% BSA in DPBS) at 4 °C overnight. After three times washing with DPBS, secondary antibodies conjugated with donkey anti-mouse IgG with Alexa 488 (1:200 Life Technologies, A-21202) and donkey anti-rabbit IgG with Alexa 568 (1:200 Life Technologies, A10042) at 1:200 dilution by blocking buffer were incubated at room temperature for 2 h. DAPI staining (Invitrogen™) was performed following the manufacturer's instructions before mounting coverslips (#1.5) with Fluoromount-G® (SouthernBiotech). The primary antibodies were mouse anti-HDAC5 (C-11) (1:50, Santa Cruz Biotechnology, sc-133225), rabbit anti-HDAC4 (1:50, Abcam, ab12174), mouse anti-FLAG (1:50, SIGMA, F1804).

**Phosphorylation analysis by LC-MS/MS.** Excised gel bands were cut into ~1-mm$^3$ pieces. The samples were reduced with 1 mM DTT for 30 min at 60 °C and then alkylated with 5 mM iodoacetamide for 15 min in the dark at room temperature. Gel pieces were then subjected to a modified in-gel chymotrypsin (Thermo Scientific) digestion procedure[101]. Gel pieces were washed and dehydrated with acetonitrile for 10 min. followed by removal of acetonitrile. Pieces were then completely dried in a speed-vac. Rehydration of the gel pieces was with 50 mM ammonium bicarbonate solution containing 12.5 ng/µl modified sequencing-grade chymotrypsin (Promega, Madison, WI) at 4 °C. Samples were then placed in a 37 °C room overnight. Peptides were later extracted by removing the ammonium bicarbonate solution, followed by one wash with a solution containing 50% acetonitrile and 1% formic acid. The extracts were then dried in a speed-vac (~1 h). The samples were then stored at 4 °C until analysis.

On the day of analysis, the samples were reconstituted in 5–10 µl of HPLC solvent A (2.5% acetonitrile, 0.1% formic acid). A nanoscale reverse-phase HPLC capillary column was created by packing 2.6-µm C18 spherical silica beads into a fused silica capillary (100-µm inner diameter × ~30-cm length) with a flame-drawn tip[102]. After equilibrating the column each sample was loaded via a Famos auto sampler (LC Packings, San Francisco CA) onto the column. A gradient was formed, and peptides were eluted with increasing concentrations of solvent B (97.5% acetonitrile, 0.1% formic acid).

As each peptide was eluted, they were subjected to electrospray ionization and then they entered into an LTQ Orbitrap Velos Pro ion-trap mass spectrometer (Thermo Fisher Scientific, San Jose, CA). Eluting peptides were detected, isolated, and fragmented to produce a tandem mass spectrum of specific fragment ions for each peptide. Peptide sequences (and hence protein identity) were determined by matching protein or translated nucleotide databases with the acquired fragmentation pattern by the software program, Sequest (ThermoFinnigan, San Jose, CA)[103]. The modification of 79.9663 mass units to serine, threonine, and tyrosine was included in the database searches to determine phosphopeptides. Phosphorylation assignments were determined by the Ascore algorithm[104]. All databases include a reversed version of all the sequences, and the data were filtered between 1 and 2% peptide false discovery rate. Supplementary Dataset 4 contains all raw LC-MS/MS data.

**FAK kinase ADP-Glo assay.** ADP-Glo kinase assay (Promega, V9101) was performed to measure FAK kinase activity. E4Y1 (poly(Glu)/poly(Tyr) ratio of 4:1) polypeptides and active FAK tyrosine kinase were also obtained from Promega (V1971). Control substrate (E4Y1), recombinant human full-length HDAC5 protein (SIGMA, SRP0107), or recombinant human full-length HDAC4 (Abnova, H00009759-P01) were incubated with active FAK with 50 µM ATP in 1× kinase

reaction buffer (Promega, V1971) at room temperature for 1 h. Then, the produced ADP was converted by ADP-Glo reagent to ATP. The amount of original consumed ATP by kinase reaction was determined with KD buffer by luminometer. FAK inhibitors (PF562271, VS-6063, PF431396) were incubated with FAK and the substrates for 1 h.

**γ-p32-ATP radioisotope FAK kinase assay.** Recombinant human HDAC5 and E4Y1 (a control substrate) were incubated with 1 µl of 10uCi/µl γ-32P-ATP (Perkin Elmer,BLU502A), 1 µl of 0.1 µg/µl FAK tyrosine kinase, 1× protease inhibitor (biochem.com), and 1 mM vanadate (NEB) in 1× kinase reaction buffer (Promega, V1971) for 1 h at room temperature. All kinase reacted protein samples were boiled with 2-mercaptoethanol containing sample buffer for 5 min and loaded into 8% polyacrylamide gels. Radioactive gels were dried overnight at room temperature using Promega gel drying kit (Promega, V7120). The dried gels were exposed to radiography films for 3 h to 24 h. All radioactive isotopes and their procedures are regulated by Radiation Safety Department at Massachusetts General Hospital.

**Data collection and analysis.** Data were collected using Windows 10, Microsoft Excel for Office 365, StepOne Software v2.3, Zeiss Zen software 2.6, Epson scan 3.9.4.7US, and Azure biosystems cSeries capture software 1.9.7.0802. The data were analyzed by Windows 10, Microsoft Excel for Office 365, Microsoft Word for Office 365, GraphPad Prism 8.4.2, StepOne Software v2.3, NIH ImageJ 1.52a, EgdeR package 3.24.1, Enrichr (https://amp.pharm.mssm.edu/Enrichr/) version January 7th, 2020, GeneXplain version 4.8, ingenuity pathway analysis (Build version: 430520 M Content version: 31813283), Overlap stats (nemates.org/MA/progs/overlap_stats.html) version 2019, Sequest (SRF v.5), and HTSeq 0.9.1.

**Statistical analyses.** All statistical analyses were performed by GraphPad Prism 9 for Windows (GraphPad Software Inc, USA). Variables were tested by either two-tailed $t$ test or Tukey test. Values were expressed as mean ± SEM unless otherwise stated. A $P$-value < 0.05 was considered significant.

**Reporting summary.** Further information on research design is available in the Nature Research Reporting Summary linked to this article.

## Data availability
All data generated or analyzed during this study are included in this published article (and its Supplementary Information Files). The RNA-seq data have been deposited to GEO under accession numbers GSE139604 and GSE144265. Source data underlying all figures are provided as a Source Data File. Source data are provided with this paper.

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

## Acknowledgements

We thank Drs. Rajeev Malhotra, Jayaraj Rajagopal, Weiguo Zou, Tatsuya Kobayashi, Amin Arnaout, Lauren Surface, and all members of the Wein laboratory for stimulating discussions. We thank Drs. Eric Olson, Jerry Feng, and Jonathan Pachter for providing reagents. We thank Dr. Hiroshi Saito for assistance with radioisotope studies, and Ross Tomaino (Taplin Mass Spectrometry Facility, Cell Biology Department, Harvard Medical School) for assistance with phosphopeptide mapping. M.N.W. acknowledges funding support from the MGH Department of Medicine (Transformative Scholars award), Paul and Cathy Braverman, the American Society of Bone and Mineral Research (Rising Star Award), the Harrington Discovery Institute, and the National Institute of Health (DK116716 and AR067285). μCT and mechanical testing were performed by the Center for Skeletal Research, an NIH-funded program (P30 AR066261). Confocal microscopy was supported by the NIH Shared Instrumentation Grant (SIG) S10OD021577. HMK acknowledges funding support from the NIH (DK011794).

## Author contributions

T.S., S.V., M.O., N.C., C.C.A., A.L., D.J.B., B.J.A., Y.U., and M.N.W. performed experiments. T.S., M.C., J.S., D.J.B., R.S., H.M.K., D.L., M.L.B., T.S.S., T.S.G., B.J.A., P.D.P., and M.N.W. analyzed the data. T.S. and M.N.W. wrote the paper. All authors reviewed and approved the paper.

## Competing interests

M.N.W. and H.M.K. receive research funding from Radius Health and Galapagos NV. D.L. has received research funding from Boehringer Ingelheim, Indalo Therapeutics and Unity Biotechnology, and has financial interests in Mediar Therapeutics and Zenon Biotech. The remaining authors declare no competing interests.
