## [Peer Review File · Nature Communications]

Reviewers' comments:

Reviewer #1 (Remarks to the Author):

What are the major claims of the paper?

The major claim of the paper is that FAK/class IIa HDAC signaling axis controls the osteocyte mechano-transduction. The paper presents in vitro and in vivo data with in depth mechanistic analyses to proof this claim. And the authors have done a very good job in presenting this claim and the story line.

Are they novel and will they be of interest to others in the community and the wider field?

In general, the claim is novel since it positions the osteocyte mechano-transduction into a core position of how cells communicate with their extracellular matrix and intracellular processes and discusses redundancies.

On a more subjective note, do you feel that the paper will influence thinking in the field?

This is a tough one – in general the mechanisms presented follow the general philosophy of the field but may also influence the future thinking.

More general comments

Based on my expertise, I have focused on the mechanical loading and its tissue consequences. The paper uses – a bit unusual for the field – a three point bending of the tibia instead of a (axial) limb loading setup. The authors should comment on this and why this is more relevant for osteocyte mechano-transduction. Limb loading (resulting in bending in the midshaft) would allow a more homogenous tissue straining across the bone cross-section. The three-point bending results in a (not controllable) maximum loading under the load tip and a gradual decrease towards the joints. Thus, osteocytes see substantially different straining in a cross-section. It is not surprising that bone formation is thus visible on the periosteal bone surface. In limb loading experiments, the endosteal bone is more reactive. To what degree is the loading set-up influencing the data and conclusions?

The data presented from the in vivo model refers to mechano-transduction results but does not identify where the samples have been taken. Since the major consequences are periosteal – how much has endosteal and mid-cortex sampling influenced the research outcomes. How have the general conclusions been influenced by such inhomogeneities? How do these link to the canaliculi-based fluid flow concept of mechano-transduction mentioned in the paper (that is not identical with the tissue strain-based model of mechanical straining). Here, the paper is not clear and not specific enough.

More Specific

A figure illustrating the loading and tissue sampling should be included to estimate/illustrate the resulting tissue straining.

axis controls the osteocyte mechanotransduction. The paper presents in vitro and in vivo data and in depth mechanistic analyses to proof this claim.

Reviewer #2 (Remarks to the Author):

This study presents results that a mechano-sensitive signaling linkage may exist involving FAK (possibly $\alpha 5 \beta 3$ integrin receptors), histone deacetylases HDAC4/HDAC5, and the tyrosine phosphorylation of HDAC5 at Y642 by FAK involved in the shuttling of HDAC5 to the nucleus and the regulation of gene expression. Although the canonical role for FAK activation in response to mechanical-based stimuli is to enhance signaling events, the authors find that fluid shear stress in

culture (6 dynes/cm) on a rotating platform causes rapid loss of FAK and adhesion protein tyrosine phosphorylation concomitant with a reduction in Sost mRNA levels (Sclerostin, involved in bone formation) in a cultured immortalized osteocyte cell line (Ocy454). Previous studies have linked pharmacological inhibition of FAK and the related Pyk2 kinases to increased bone density, but the cell targets and mechanisms remain unknown.

The authors make the conclusion that FAK signaling in “static” conditions may be facilitating Sost expression. Pharmacological or CRISPR knockout of FAK reduce Sost mRNA levels in vitro. However, causality is weak, the relevance of HDAC5 Y642 phosphorylation is not shown, and there are concerns that rapid loss of FAK tyrosine phosphorylation in the fluid flow shear stress (FFSS) assay may be a technical artifact. Several studies using bone and endothelial cells have linked low levels of FFSS (12 dynes/cm or less) to FAK and signal pathway activation. This osteocyte cell line has unique properties in vitro. It remains unclear whether the in vitro and in vivo findings are comparable. The results are structured around loss of HDAC or FAK signaling. This study would have much greater impact to follow a stimulated activity and determine the effects of a specific post-translational modification of HDAC5 function.

Suggested improvements:

1. FAK signaling is tightly controlled. FAK Y397 phosphorylation is just one indirect measure. What differences occur between 2D and 3D culturing? Can the authors define what is occurring for peripheral adhesion changes in short versus long-term FFSS? the platform cell rotation assay to achieve FFSS is very low tech and forces will not be the same across the dish of cells.
2. Line 3, page 2. No evidence is presented that FAK is “transiently” suppressed by FFSS. The longest time point for biochemical experiments was 90 min (still reduced FAK Y397).
3. Line 5, page 2. There is no evidence that HDACs are serving as “sensors”.
4. If rapid FAK and paxillin tyrosine dephosphorylation is a mechano-response, then it is likely that another receptor (phosphatase) is serving as an upstream mechano-sensor.
5. Sclerostin is known to inhibit Wnt signaling. FAK signaling facilitates Wnt/b-catenin signaling in development and in pathologies such as cancer. The results presented herein are opposite signaling linkage and need better explanation and proof.
6. Other measures of shear stress need to be incorporated into this study. The impact of FFSS on osteocytes (normally dispersed within bone) remains controversial.
7. page 4, line 10: “Here we show that FAK regulates class IIa HDAC subcellular localization dt direct tyrosine phosphorylation.” This statement remain unsupported as no analyses were performed to evaluate the effects of HDAC5 Y642F mutation in an appropriate cell background.
8. Fig. 1E. The non-phospho “active” b-catenin staining is not controlled, total b-catenin staining needs to be verified, the cells stained need to be elucidated, and multiple images quantified. How does this fit with a H4H5 DKO signaling linkage - indirectly via changes in Sost levels?
9. Fig. 2B. The images shown for HDAC4 or HDAC5 nuclear accumulation by indirect IF cell staining is weak and the images shown are not convincing. This is an important part of this paper and a secondary method of measuring nuclear versus cytoplasmic distribution is needed.
10. It is published that the Ocy454 cells express GFP from a DMP1 promoter that was used in the creation-selection. How are the authors able to obtain IF images using 488 nm dyes?
11. Fig. 2F and G. Better methods and statistical analyses are needed for the RNAseq data shown throughout the paper. Ddo the changes represent increases, decreases, or just change from

control?

12: Supplemental Figure 2. Legend needs additional information. Fig. 2B. what cells are being used? Are these the H4H5DKO? Can the authors show re-distribution of flag-tagged HDAC5 in response to FFSS? If HDAC5 S2A is constitutively localized, what is the status and role for HDAC5 tyrosine phosphorylation. How can the authors find such a large change in the flag-tagged HDAC5 transfected cells for Sost when it seems that less than 50% of cells have been transfected?

13. Fig. 3C. Images of HDAC4 or HDAC5 nuclear accumulation in response to FFSS are not convincing. Need cell fractionation.

14. Fig. 4. Authors need to better characterize CRISPR knockout of FAK in Oct454 cells. At least two clones (sequence verified) need to be analyzed. At a minimum, FAK should be re-expressed to prove causality. As presented, the results cannot be interpreted effectively.

15. RNAseq data. The minimum number of biological replicates for RNAseq experiments is three. It is unclear whether RPKM counts under a certain level were included. The read depth is low (20 M) and the quality of sequence data generated uncertain since data was not deposited for review.

16. Fig. 4C and D. these results are ambiguous. The level of Sost mRNA is already very low in the FAK-KO cells. It is unclear how the authors can conclude that FAK pathway inhibition does not alter PTH Sost suppression.

17. Fig. 4E-G. The volcano and heat maps generated for the FAK-KO cells are odd and difficult to interpret.

18. Fig. 4. Is the FAK-related protein Pyk2 not involved in the proposed regulation. Is it expressed in the Ocy454 cells?

19. Fig. 5A. This rapid dephosphorylation of FAK and paxillin is difficult to understand.

20. Fig. 5. The p value for Sost change with FAK inhibition is quite small and the small overlap in Venn identified targets (not listed) between FAK-KO and FAK inhibitor is worrisome and suggest weakness in the model system.

Reviewer #3 (Remarks to the Author):

In this manuscript Sato et al provide a mechanistic link between mechanical stimulation of osteocytes and suppression of Sost, a repressor of bone formation/regeneration and target of a new osteoporosis drug. This report provides evidence that shear stresses induced by fluid flow (FFSS) across osteocyte membranes reduces serine phosphorylation of Hdac4 and Hdac5, thereby facilitating their nuclear localization or retention, their interactions with Mef2C at the Sost promoter, and repression of Sost gene transcription. Because serine phosphorylation of Hdac4/5 was partially but not completely suppressed by FFSS, the authors hypothesized that other pathways were also involved. RNA sequencing studies led the authors to the focal adhesion signaling pathway. The authors focused on one candidate, FAK, and embarked on a series of nice experiments to test its role in Sost expression. The authors showed that FFSS disrupts constitutive FAK signaling caused by integrin attachments to the extracellular matrix. Consequently FAK tyrosine phosphorylation of Hdac5 is reduced in osteocytes and Sost transcription is repressed.

As a whole, the work is conceptually novel for the bone biology community and the newer mechanobiology sub-community within it. The use of in vivo and in vitro models of mechanical loading/FFSS to study signaling and Sost expression is a strength. The work will likely gain

attention from a wider field because of the discovery that Hdac5 is phosphorylated by FAK on a tyrosine residue and a new pY-Hdac5 antibody was generated. The manuscript is compelling but additional evidence is needed to strengthen the conclusions.

1. The conclusions (and title) state that FFSS and FAK control class IIa Hdac signaling. This is too broad as not all class IIa Hdacs were studied. Hdac5 was studied in greatest depth and Hdac4 was assessed in many but not all experiments. There is no mention of Hdac7 or Hdac9 in the text. The RNA-seq data provided in the supplemental table indicate that Hdac7 has similarities to Hdac5 in that it is highly expressed in osteocytes and is down regulated by FFSS and in FAK KO cells. It would be ideal to at least mention potential roles Hdac7 and Hdac9 in the manuscript, perhaps in the Discussion. The title and conclusions should be adjusted as well, focusing on just Hdac4/5 (or perhaps just Hdac5 as the comments below are taken into consideration).

2. More rationale is needed for the mouse models used in the first two figures and the first two supplemental figures. One issue that needs to be explained is why Hdac4 cKO mice and Hdac5 KO mice were used for these studies. The experiments would be more convincing if Hdac5 cKO mice (Dmp1-Cre) were used to show the cell-autonomous role of Hdac5 in osteoblasts/osteocytes. The use of mixed cKO and KO mice also raises an important issue about controls, which appear to be wildtype and not expressing Dmp1-Cre+. At the very least, the limitations of these models needs to be addressed.

3. Several panels within Figures 1 and 2 and Supplemental Fig 1/2 only show two or three of the four groups needed. For example, only the control and DKO mice are shown in Fig 1B-D. The KO and cKO mice should be added to these figures or to supplemental data. In Figure 2A and supplemental Fig 1 (ABC), the Hdac4 cKO mice are not shown.

4. Additional details are needed on methods for the loading and FFSS experiments. What is the age of the mice used for short-term and long-term mechanical loading (p16, line 24)? When were tissues and mRNA isolated? Were they immediately isolated after loading/FFSS stopped or did some time lapse between loading/FFSS and fixation/cell lysis?

5. In Figure 2B, the Hdac4 looks mostly cytoplasmic in Ocy454 cells after FFSS, contrary to the conclusions. The densitometric studies on Hdac4 and Hdac5 should be verified with western blots of nuclear and cytoplasmic cell fractions.

6. The results in Figure 2D are interesting and puzzling. How does Hdac4/5 deficiency prevent Mef2C binding to DNA? Mef2C should be recruiting Hdac4/5 to that region of DNA, not the other way around. Are Mef2C levels in the nucleus different in the DKO cells?

7. The terms "monoclonal" and "polyclonal" are used on several occasions to distinguish cell lines generated by single-cell cloning from bulk populations transduced with viruses. In immunology, monoclonal is used to describe the unique Ig or TCR sequences formed by DNA recombination. There is no equivalent in osteoblasts/osteocytes. I'm not sure what monoclonal means for virally-transduced cells. Single cell subclones is a better and more traditional description of what was done here.

8. It says that 6 "monoclonal" FAK KO lines were generated (page 20), but it is not clear in the methods what the cell of origin is. Is it Ocy454? Western blots are shown for only 2 in Figure 3B. It appears that 2 clones were used to generate data in Fig 4G too, but this needs clarity. Overall, it is fine to focus on subclones, but there should be rationale for why they were picked, they should be named and used consistently throughout the study.

9. In Fig 5, were cells treated with FAK inhibitors also LV-transduced and selected? Or were just wildtype cells treated with the inhibitors? Potential effects of LV-transduction on transcriptomic results could be added to an otherwise good discussion on page 9, lines 12-20.

10. To place the results into biological context, it would be essential to know what FAK inhibitors do to Ocy454 viability and function at the concentrations tested.

11. The role of pY on Hdac4 and Hdac5 nuclear localization deserves more analysis and necessary support to support the conclusions (page 11, line 5-7) and models (Fig 8E) proposed. The IF studies in Supp. Fig 4b are insufficient. Examining how pY functions in cooperation with pS would be interesting but not necessary for this study.

12. The western blot in Figure 6J indicates that Hdac5 levels were higher in FAK KO and increase further following with FFSS. Hdac5 transcript levels also appear to be higher in the RNA-seq studies. Please discuss the potential meaning of this.

13. How does cilengitide affect pY-Hdac4/5 and pS-Hdac4/5, as well as Hdac4 and Hdac5 overall levels? The addition of these western blots to Figure 7BC would link these experiments to the rest of the study better.

14. The FAK inhibitor in vivo experiments in Fig 8 are a reasonable approach in the absence of FAK cKO (Dmp1-cre), but the results in Fig 8a from liver extracts are not convincing.

15. In the Discussion, lines 16-17 on page 13 contradict what is stated in lines 1-2 of page 14. The latter is more accurate unless experiments requested above on how pY affects Hdac4/5 nuclear localization can be provided.

Minor comments:

1. On page 2, line 5, HDACs are called "sensors". This was not shown. Better terms would be transducers, facilitators or amplifiers of mechanical signals and cell membrane interactions.

2. In Figure 1A, Hdac4 cKO is abbreviated H4KO in the graphs. This is misleading and should be changed to Hdac4 cKO. Ditto for Supplemental figure 2A.

3. Please clarify which cells were transfected with Hdac5 expression vectors in the legend of supplemental figure 2b. The methods section indicates that it was the Hdac5 KO cells, but Ocy454 cells are mentioned in the legend for panel C so it is a bit confusing as to what is shown in Suppl Fig 2b.

4. Page 7, line 2: Please clarify what "rapid" describes or omit this word. 3 hours doesn't seem "rapid" for signaling studies.

5. It would be helpful to define what is meant by the term "tonic" signaling when the term is used for the first time. It may be unfamiliar to some in the bone community. It appears to refer to constitutive signaling/kinase activity in the basal state.

We found the reviewers' comments to be extremely helpful, and have performed multiple additional experiments and analyses which we believe strengthen the validity and impact of our findings. Our point-by-point responses (in red) to the specific concerns raised by each reviewer (in black) are found below. In the revised manuscript text, all changes are highlighted in yellow.

Reviewer #1 (Remarks to the Author):

What are the major claims of the paper?

The major claim of the paper is that FAK/class IIa HDAC signaling axis controls the osteocyte mechano-transduction. The paper presents in vitro and in vivo data with in depth mechanistic analyses to proof this claim. And the authors have done a very good job in presenting this claim and the story line.

Are they novel and will they be of interest to others in the community and the wider field?

In general, the claim is novel since it positions the osteocyte mechano-transduction into a core position of how cells communicate with their extracellular matrix and intracellular processes and discusses redundancies.

On a more subjective note, do you feel that the paper will influence thinking in the field?

This is a tough one – in general the mechanisms presented follow the general philosophy of the field but may also influence the future thinking.

We thank the reviewer for comments indicating that our manuscript is clearly-written, novel, and will influence future thinking in the field of skeletal mechano-transduction.

More general comments

1. Based on my expertise, I have focused on the mechanical loading and its tissue consequences. The paper uses – a bit unusual for the field – a three point bending of the tibia instead of a (axial) limb loading setup. The authors should comment on this and why this is more relevant for osteocyte mechano-transduction. Limb loading (resulting in bending in the midshaft) would allow a more homogenous tissue straining across the bone cross-section. The three-point bending results in a (not controllable) maximum loading under the load tip and a gradual decrease towards the joints. Thus, osteocytes see substantially different straining in a cross-section. It is not surprising that bone formation is thus visible on the periosteal bone surface. In limb loading experiments, the endosteal bone is more reactive. *To what degree is the loading set-up influencing the data and conclusions?*

We thank the reviewer for this important question. As noted, the axial tibia compression model has been widely adopted and possesses a number of strengths, including reproducibility across groups, ease of implementation, and the ability to explore trabecular bone adaptation. However, we implemented the cantilever bending cortical bone model in this study as its strengths (noted below) aligned closely with our objectives. We have clarified the rationale for our choice of exogenous loading model in the context of our discussion (page 14 lines 23-25, page 15 lines 1-2). As well, we have clarified the strain environment induced at the diaphyseal cross-section where dynamic histomorphometry was performed (please see **new Supplemental Figure 1**).

Our previous studies have demonstrated that the mid-shaft normal strain distribution induced by our model of cantilever bending (not 3-pt bending) closely mirrors that induced by locomotion (1, 2). Further, due to tibial morphology, peak normal strains within a 3 mm region of the tibia mid-shaft are fairly constant. As a result, dynamic histomorphometry outcomes within this region are also constant. Although the alternative axial tibia compression model induces substantial periosteal bone formation (3, 4), investigators have focused on the endocortical surface to avoid potential confounding of mechanotransduction-induced responses with woven bone injury responses.

Phenotypically, our cantilever bending model demonstrates modest heterogeneous periosteal osteoblast activation at loading and strain magnitudes much lower than those implemented in the axial tibia compression model. This adaptive response is reminiscent of the modest periosteal bone formation induced in humans by exercise interventions and is consistent with current bone mechanotransduction paradigms (i.e., osteocyte mechanosensing translated into a periosteal bone formation response). We have used this model to identify a number of novel bone mechanotransduction insights that have been subsequently confirmed by other groups using other models (e.g., rest inserted loading (5)). Our choice of model and analysis was therefore based on our hypothesis that the FAK/HDAC4/5 signaling axis controls bone mechanotransduction. Our *in vivo* results (downregulation of osteocyte sclerostin in loaded WT, but loss of this effect in *Hdac4/5* DKO, in parallel with the loss of load induced periosteal osteoblastic response, Fig 1a and b) support the experimental approach used here.

2. The data presented from the *in vivo* model refers to mechano-transduction results but does not identify where the samples have been taken. Need to clarify text, align images in figure. Since the major consequences are periosteal – how much has endosteal and mid-cortex sampling influenced the research outcomes. How have the general conclusions been influenced by such inhomogeneities? How do these link to the canaliculi-based fluid flow concept of mechano-transduction mentioned in the paper (that is not identical with the tissue strain-based model of mechanical straining). Here, the paper is not clear and not specific enough.

Regarding the sampling location and induced strain environment, please see response #1 and **new Supplemental Figure 1**. The response of the tibia to cantilever bending is primarily periosteal at the magnitude of external loading implemented in this study. As noted above, this adaptive response is consistent with the literature regarding osteocytes serving as the primary cellular mechanotransducers and communicating with surface lining cells to drive required bone formation. In other models, at much greater strain magnitudes than induced in this study, groups have identified activity on both periosteal and endocortical surfaces.

Additionally, we thank the reviewers for identifying the lack of clarity connecting canaliculi-based fluid flow with the model of cantilever bending-induced tissue strain. To address this shortcoming, we have added a functional description of how intracortical fluid flow is achieved

through gradients in the mechanical strain environment of bone (page 3, final paragraph), and clarified how the implemented model of cantilever bending provides a broad testing platform for exploring *in vivo* cellular mechanosensing (canaliculi-based fluid flow being one potential principal mechanism; page 14-15)

More Specific

A figure illustrating the loading and tissue sampling should be included to estimate/illustrate the resulting tissue straining.

In response to this comment, we have generated Supplemental Figure 1 which illustrates the loading orientation of the tibia, the mid-diaphysis sampling site, and the normal strains induced around that cross-section of bone.

The manuscript currently mixes the mechanical stimulus that comes from limb loading and the one that comes from fluid flow and leads to control of osteocyte mechanotransduction. These mechanical stimuli, however, are different. The paper presents *in vitro* and *in vivo* data on in depth mechanistic analyses of mechanotransduction to proof this claim but mixes these mechanical stimuli and their effects on tissues (limb loading leads to osteoblastic and osteoclastic activity at the endost or periost bone surface; fluid flow around osteocytes would lead to alterations in bone mineral within the cortex). Please clarify which effect you concentrate on.

Please see response to question #2 above.

Reviewer #2 (Remarks to the Author):

This study presents results that a mechano-sensitive signaling linkage may exist involving FAK (possibly avb3 integrin receptors), histone deacetylases HDAC4/HDAC5, and the tyrosine phosphorylation of HDAC5 at Y642 by FAK involved in the shuttling of HDAC5 to the nucleus and the regulation of gene expression. Although the canonical role for FAK activation in response to mechanical-based stimuli is to enhance signaling events, the authors find that fluid shear stress in culture (6 dynes/cm) on a rotating platform causes rapid loss of FAK and adhesion protein tyrosine phosphorylation concomitant with a reduction in *Sost* mRNA levels (*Sclerostin*, involved in bone formation) in a cultured immortalized osteocyte cell line (Ocy454). Previous studies have linked pharmacological inhibition of FAK and the related Pyk2 kinases to increased bone density, but the cell targets and mechanisms remain unknown.

The authors make the conclusion that FAK signaling in “static” conditions may be facilitating *Sost* expression. Pharmacological or CRISPR knockout of FAK reduce *Sost* mRNA levels *in vitro*. However, causality is weak, the relevance of HDAC5 Y642 phosphorylation is not shown, and there are concerns that rapid loss of FAK tyrosine phosphorylation in the fluid flow shear stress (FFSS) assay may be a technical artifact. Several studies using bone and endothelial cells have linked low levels of FFSS (12 dynes/cm or less) to FAK and signal pathway activation. This osteocyte cell line has unique properties *in vitro*. It remains unclear whether the *in vitro* and *in vivo* findings are comparable. The results are structured around loss of HDAC or FAK signaling. This study would have much greater impact to follow a stimulated activity and determine the effects of a specific post-translational modification of HDAC5 function.

Suggested improvements:

1. FAK signaling is tightly controlled. FAK Y397 phosphorylation is just one indirect measure. What differences occur between 2D and 3D culturing? Can the authors define what is occurring

for peripheral adhesion changes in short versus long-term FFSS? the platform cell rotation assay to achieve FFSS is very low tech and forces will not be the same across the dish of cells.

We thank the reviewer for these important comments. We agree that the platform cell rotation assay is a “low tech” method to achieve FFSS in cultured Ocy454 cells. For this reason, we have performed FFSS experiments using a laminar flow chamber (6). Similar to our results in the platform cell rotation model, we observed that short-term exposure of Ocy454 cells to a uniform and well-defined “dose” (8 dynes/cm²) of FFSS leads to reduced FAK Y397 and HDAC5 Y642 phosphorylation (**new Supplemental Figure 7G**).

In addition, we have performed long term FFSS experiments in our cell rotation model. In these studies, we note that 8.5 and 17 hours of FFSS treatment do not lead to reductions in FAK and HDAC5 tyrosine phosphorylation (**Figure R1**). We attempted to assess the effects of 3D culture on Ocy454 cell FAK phosphorylation using a 3D collagen gel system. Unfortunately, the enzymatic digestions required to liberate cells from this 3D environment precluded our ability to detect FAK and HDAC5 phosphorylation.

Regarding the point that FAK Y397 phosphorylation is just one indirect measure of FAK activity, here we note that we also assessed phosphorylation of paxillin at Y118, a generally-accepted marker of FAK activity in cells. Indeed, FFSS treatment and FAK gene deletion reduces Paxillin Y118 phosphorylation as well (Figures 4A and 3B).

Figure R1. Ocy454 cells were treated plus/minus FFSS for 8.5 hours (A) and 17 hours (B), following by immunoblotting.

2. Line 3, page 2. No evidence is presented that FAK is “transiently” suppressed by FFSS. The longest time point for biochemical experiments was 90 min (still reduced FAK Y397).

We agree with this point, and have changed the word “transiently” on page 2 line 6 to “rapidly”. As noted above, Figure R1 shows the lack of effects of prolonged FFSS on FAK and HDAC5 tyrosine phosphorylation.

3. Line 5, page 2. There is no evidence that HDACs are serving as “sensors”.

We agree with this comment. Here, the word “sensors” was used to indicate that class IIa HDAC subcellular localization changes in response to sensing FFSS. We by no means meant to imply that HDAC5 and HDAC4 directly sense mechanical cues. To clarify this matter, the word “sensors” has been changed to “transducers.”

4. If rapid FAK and paxillin tyrosine dephosphorylation is a mechano-response, then it is likely that another receptor (phosphatase) is serving as an upstream mechano-sensor.

The nature of the upstream mechanosensory responsible for mediating changes in FAK activity in response to FFSS remains to be precisely defined. Our data in Figure 7 demonstrates that a cellingitide-sensitive integrin heterodimer controls FAK/HDAC5 phosphorylation and SOST expression. Future work is needed to identify whether active dephosphorylation by protein phosphatases triggered by FFSS is also involved in osteocyte mechanotransduction. Here, we note that treatment of Ocy454 cells with vanadate (a non-specific inhibitor of tyrosine phosphatases) increases HDAC5 Y642 and FAK Y397 phosphorylation in cells (**Figure R2**). These data indicate that dynamic tyrosine phosphorylation and dephosphorylation of these proteins is always occurring in osteocytes, providing a potential explanation for why acute inhibition of FAK could lead to rapid HDAC5 Y642 dephosphorylation. Please see point #19 below for further discussion.

Figure R2. Ocy454 cells were treated plus/minus vanadate (1 mM) for 60 minutes followed by immunoblotting.

5. Sclerostin is known to inhibit Wnt signaling. FAK signaling facilitates Wnt/b-catenin signaling in development and in pathologies such as cancer. The results presented herein are opposite signaling linkage and need better explanation and proof.

We thank the reviewer for bringing up the literature linking FAK signaling to WNT pathway activation in certain cancers. Here, we note that regulation of WNT pathway output is highly dependent on cellular context. In osteocytes, the balance between secretion of WNT ligands and WNT inhibitors controls cellular WNT activity. SOST is an osteocyte-specific WNT inhibitor that potently controls osteocyte WNT activity in an autocrine/paracrine manner (7). Our data indicate that FAK signaling increases SOST expression. Here, we would like to highlight **new data in Supplemental Figure 5C** that FAK-deficient cells reconstituted with FAK which confirms this model.

6. Other measures of shear stress need to be incorporated into this study. The impact of FFSS on osteocytes (normally dispersed within bone) remains controversial.

As described above in point #1, we now have performed FFSS experiments in laminar flow chambers at 8 dynes/cm² which confirm our model. These data are shown in **new Supplemental Figure 7G**.

7. page 4, line 10: "Here we show that FAK regulates class IIa HDAC subcellular localization dt direct tyrosine phosphorylation." This statement remain unsupported as no analyses were performed to evaluate the effects of HDAC5 Y642F mutation in an appropriate cell background.

We thank the reviewer for raising this extremely important point. In response to this question, we performed many additional experiments to assess the function of HDAC5 Y642 using the HDAC5^{Y642F} point mutant which was re-introduced into HDAC5-deficient Ocy454 cells using lentiviral-mediated gene transfer (**new Figure 6E**). These data demonstrate that:

1. HDAC5^{Y642F} shows increased nuclear localization in Ocy454 cells as assessed by subcellular fractionation (**new Figure 6F**)
2. HDAC5^{Y642F} shows increased nuclear localization in Ocy454 cells as assessed by immunocytochemistry (**new Figure 6G**)

3. HDAC5^{Y642F}-reconstituted cells show reduced SOST expression compared to HDAC5 WT expressing cells (new Figure 6H)

8. Fig. 1E. The non-phospho “active” b-catenin staining is not controlled, total b-catenin staining needs to be verified, the cells stained need to be elucidated, and multiple images quantified. How does this fit with a H4H5 DKO signaling linkage - indirectly via changes in Sost levels?

This is an important point. We have quantified the immunohistochemistry stains and this quantification is now included in a **revised Figure 1F**. The second question here is also very important: HDAC4/5 deficiency increases sclerostin expression and thereby reduces WNT activity. We have added a sentence on page 5 (line 25-26) to explain this linkage: “Sclerostin down-regulation in osteocytes is needed for loading-induced WNT pathway activation and osteoblastic bone formation (8, 9).”

9. Fig. 2B. The images shown for HDAC4 or HDAC5 nuclear accumulation by indirect IF cell staining is weak and the images shown are not convincing. This is an important part of this paper and a secondary method of measuring nuclear versus cytoplasmic distribution is needed.

This is an excellent point. To confirm the immunocytochemistry results, we have performed subcellular fractionation to measure the localization of HDAC4 and HDAC5 in response to FFSS and FAK inhibitor treatment. These data are shown in **new Figure 2D**. Overall, these results confirm the initial data shown using immunofluorescence and strengthen our claim that FFSS and FAK inhibitors promote the nuclear translocation of HDAC4 and HDAC5.

10. It is published that the Ocy454 cells express GFP from a DMP1 promoter that was used in the creation-selection. How are the authors able to obtain IF images using 488 nm dyes?

This is correct: Ocy454 cells were initially obtained from DMP1-GFP^{topaz} expressing mice. However, in this single cell clone derived from the initial Ocy454 cell population, DMP1-GFP^{topaz} expression decreases over time in culture such that very little/no GFP fluorescence is noted in un-transfected cells under our growth conditions.

11. Fig. 2F and G. Better methods and statistical analyses are needed for the RNAseq data shown throughout the paper. Do the changes represent increases, decreases, or just change from control?

We have worked extensively with co-authors Ruslan Sadreyev (director the MGH Molecular Biology bioinformatics core) and Murat Cetinbas to ensure that all RNAseq data have been analyzed appropriately using state-of-the-art statistical methods. Changes represent fold change values versus control within each experiment. All figure legends, methods, and text have been reviewed and edited for clarity. To answer this specific question: for Figure 2G, we have clarified the corresponding figure legend to state “In this graph, the x-axis corresponds to the gene ontology enrichment score for genes differentially-expressed in response to FFSS.” For Figure 2H, we similarly have clarified the figure legend to note “In this graph, the x-axis corresponds to the upstream analysis score (adjusted p value) for genes that were differentially-expressed in response to FFSS.” Further details regarding RNAseq data analysis and representation are found below.

12: Supplemental Figure 2. Legend needs additional information. Fig. 2B. what cells are being used? Are these the H4H5DKO? Can the authors show re-distribution of flag-tagged HDAC5 in response to FFSS? If HDAC5 S2A is constitutively localized, what is the status and role for

HDAC5 tyrosine phosphorylation. How can the authors find such a large change in the flag-tagged HDAC5 transfected cells for Sost when it seems that less than 50% of cells have been transfected?

We thank the reviewer for the opportunity to clarify these important points. Due to inclusion of new data, previous Supplemental Figure 2 is now Supplemental Figure 3 in the revised manuscript. For Supplemental Figure 3B, HDAC5-deficient cells were reconstituted with lentiviral constructs expressing FLAG-tagged wild type (WT) and S259/498A (2SA) mutants. The figure legend for this supplemental figure has been revised to clarify this point. We have examined the localization of both endogenous and FLAG-tagged lentiviral delivered HDAC5 in Ocy454 cells in response to FFSS. In Figure 2B, we note that FLAG-tagged HDAC5 translocates from the cytoplasm to the nucleus in response to FFSS. In Supplemental Figure 4B, we report the localization of *endogenous* HDAC5 in Ocy454 cells in response to FFSS. Endogenous and lentiviral-expressed FLAG-HDAC5 show similar basal subcellular localization patterns, and both proteins translocate into the nucleus in response to FFSS.

The comment here about the role of tyrosine phosphorylation of HDAC5 in the 2SA mutant is quite interesting. Clearly the 2SA mutant shows striking nuclear translocation, consistent with previous literature on the role of S259 and S498 phosphorylation in the subcellular localization of HDAC5. Our new data indicates that HDAC5 Y642 also plays a role in controlling the subcellular localization of this protein (see point #7 above, new data in Figure 6E-H). Future studies are needed to determine the relationship between tyrosine 642 phosphorylation by FAK and serine phosphorylation by salt inducible kinases. The strong phenotype of the HDAC5 2SA mutant indicates that this construct is not the best tool to address this question.

Regarding the final question about the transduction efficiency of the HDAC5 2SA mutant, here we note that cells were infected with these lentiviral particles and then were selected with puromycin such that only puromycin-resistant cells were analyzed in these experiments in Supplemental Figure 3B. Certainly, variable expression levels of the HDAC5 2SA construct are possible in this experimental design. The thresholds used to detect Alexa 488 signals here may be insufficient to visualize cells expressing low levels of FLAG-HDAC5 2SA.

13. Fig. 3C. Images of HDAC4 or HDAC5 nuclear accumulation in response to FFSS are not convincing. Need cell fractionation.

We thank the reviewer for this important point. In response to this comment, we have performed additional experiments where we investigated the localization of HDAC4 and HDAC5 by subcellular fractionation. As shown in **new Figure 2D**, this approach confirms our microscopy data and clearly demonstrates that these two factors translocate into the nuclear in response to FFSS.

14. Fig. 4. Authors need to better characterize CRISPR knockout of FAK in Oct454 cells. At least two clones (sequence verified) need to be analyzed. At a minimum, FAK should be re-expressed to prove causality. As presented, the results cannot be interpreted effectively.

This also is a very important point. We have used multiple independent approaches to determine the role of FAK in FFSS signaling, including pharmacologic inhibitors and CRISPR loss of function data. In response to this point, we have 'rescued' our FAK-deficient single cell clone 1-4 with a lentiviral construct expressing FAK. As expected, reconstitution with FAK rescues defects in SOST expression (**new Supplemental Figure 5C**). These results add

confidence to our model that FAK plays a key role in controlling HDAC5 localization and SOST expression.

15. RNAseq data. The minimum number of biological replicates for RNAseq experiments is three. It is unclear whether RPKM counts under a certain level were included. The read depth is low (20 M) and the quality of sequence data generated uncertain since data was not deposited for review.

Based on extensive discussions with co-authors Drs. Sadreyev and Cetinbas, we have clarified RNAseq data analysis and presentation in this revised manuscript. Fastq files have been uploaded to GEO under accession numbers GSE139604 and GSE144265. Generally, 20-30 million reads per sample is a typical recommended range for standard cell culture RNAseq studies. As a basic standard in RNA-seq analyses, in each comparison between groups of samples, we removed genes with low expression level. In particular, genes that had count per million reads (CPM) above 1.0 in less than two samples were removed from the analysis. **New Supplemental Table 3** shows the quality control measures (read numbers, mapping frequencies) for each RNAseq library that was sequenced, this information has also been uploaded to GEO with the submission package. Regarding the number of replicates 'required' for RNAseq studies, here we note that we performed multiple independent control experiments where "static" Ocy454 cells were analyzed. A principal component plot merging all data is shown in new Supplemental Figure 6B. As noted, most control samples are relatively close in PC space, indicating robust data quality. 2-3 replicates for each biologic condition were performed in our experiments. We agree that large numbers of replicates are needed for RNAseq studies when heterogeneous primary tissues are analyzed. However, since replicates in these cell culture experiments are generally similar to each other, we feel that this approach is sufficiently robust to support the claims here regarding global patterns of gene expression changes in response to FFSS.

16. Fig. 4C and D. these results are ambiguous. The level of Sost mRNA is already very low in the FAK-KO cells. It is unclear how the authors can conclude that FAK pathway inhibition does not alter PTH Sost suppression.

We thank the reviewer for the opportunity to clarify this important point. Although SOST expression is low in FAK-deficient cells, there certainly is detectable expression. These low levels of SOST are not further decreased by FFSS. However, PTH treatment clearly reduces SOST expression in FAK-deficient cells, demonstrating that PTH-mediated SOST suppression is independent of FAK, and that our models are capable of distinguishing these low levels of expression. In revision, we have confirmed these data which are shown in **revised Figures 3C, D** (please note that figure numbering has changed to accommodate incorporation of new data) which now also shows absolute values of SOST mRNA (relative to beta-actin).

17. Fig. 4E-G. The volcano and heat maps generated for the FAK-KO cells are odd and difficult to interpret.

The legend for Figure 3E has been revised to state: "Volcano plots representing the effects of FFSS in each cell type (control and FAK-KO) are shown. In these plots, each data point represents a distinct gene. Differentially-expressed genes (DEGs) ($\log_2FC > 1$ or < -1 , $FDR < 0.05$) are shown as blue data points. While many DEGs are noted in response to FFSS in control cells, very few FFSS-induced DEGs are observed in FAK-KO cells." We believe that these plots are an impactful way to demonstrate transcriptomic effects of FFSS in control cells, and the relative lack of such effects of FFSS in FAK-KO cells. For the heat map shown in Figure

3G, we have revised the methods section to clearly explain how this heat map was generated, and the figure legend has been revised accordingly.

18. Fig. 4. Is the FAK-related protein Pyk2 not involved in the proposed regulation. Is it expressed in the Ocy454 cells?

Based on our RNAseq results, expression of FAK is at least 20-fold higher than Pyk2 in Ocy454 cells (**Figure R3**). Moreover, we note a clear phenotype observed when FAK alone is deleted. As shown in Supplemental Figures 5A and 5B, bulk cell populations expressing sgRNAs targeting PYK2 showed normal FFSS-mediated SOST suppression.

Figure R3. RNAseq data from Ocy454 cells grown at 37°C for 14 days showing RPKM counts for FAK (PTK2) and PYK2.

19. Fig. 5A. This rapid dephosphorylation of FAK and paxillin is difficult to understand.

Related to point #4 above (see Figure R2), we note that treatment of Ocy454 cells with the phosphatase inhibitor vanadate leads to increases in FAK substrate phosphorylation. This indicates that FAK substrates are subjected to rapid phosphorylation and dephosphorylation under steady state conditions. As such, cellular perturbations that only block FAK kinase activity have the potential to lead to quick reductions in FAK substrate phosphorylation levels due to high constitutive cellular phosphatase activity. Rapid “on/off” signaling via this mechanism is thought to contribute to robust cellular signaling output (for review, (10)).

20. Fig. 5. The p value for Sost change with FAK inhibition is quite small and the small overlap in Venn identified targets (not listed) between FAK-KO and FAK inhibitor is worrisome and suggest weakness in the model system.

We thank the reviewer for raising this important point. As shown in Figures 3A, we note robust reductions in SOST expression in response to multiple pharmacologic FAK inhibitors. Furthermore, in the RNAseq dataset shown in Figure 4D, the FDR (p value adjusted for multiple comparisons) is indeed quite small at 1.56×10^{-8} . This data point on the Volcano plot is relatively “low” on the y-axis since some of the other changes seen in response to PF562271 had even smaller p values. Nonetheless, these RT-qPCR and RNAseq data clearly demonstrate that SOST expression is reduced by FAK inhibitor treatment.

Regarding the second point made in this comment, the Venn diagram in Figure 4E shows significant overlap between the acute effects of FAK inhibitors and the chronic effects seen when FAK is deleted in Ocy454 cells. We certainly were not expecting to observe ‘perfect’ overlap between the effects of these distinct perturbation. Chronic FAK deletion may lead to secondary gene expression changes in the genetic system, and acute treatment with PF562271 may lead to changes in gene expression related to inhibition of cellular kinases other than FAK. Nonetheless, the overlap noted between these two manipulations is highly statistically significant, as we have noted in the revised manuscript text: “First, 200 of the 975 genes regulated by PF562271 were also regulated by FAK gene deletion ($p < 4.987 \times 10^{-13}$ for the statistical significance of the overlap between two groups), indicating largely concordant effects between acute pharmacologic and chronic genetic FAK inhibition.”

In addition, we have now provided a separate tab on Supplemental Table 1 listing the DEGs noted for each RNAseq comparison used for Figure 4E (Venn diagrams).

Reviewer #3 (Remarks to the Author):

In this manuscript Sato et al provide a mechanistic link between mechanical stimulation of osteocytes and suppression of *Sost*, a repressor of bone formation/regeneration and target of a new osteoporosis drug. This report provides evidence that shear stresses induced by fluid flow (FFSS) across osteocyte membranes reduces serine phosphorylation of Hdac4 and Hdac5, thereby facilitating their nuclear localization or retention, their interactions with Mef2C at the *Sost* promoter, and repression of *Sost* gene transcription. Because serine phosphorylation of Hdac4/5 was partially but not completely suppressed by FFSS, the authors hypothesized that other pathways were also involved. RNA sequencing studies led the authors to the focal adhesion signaling pathway. The authors focused on one candidate, FAK, and embarked on a series of nice experiments to test its role in *Sost* expression. The authors showed that FFSS disrupts constitutive FAK signaling caused by integrin attachments to the extracellular matrix. Consequently FAK tyrosine phosphorylation of Hdac5 is reduced in osteocytes and *Sost* transcription is repressed.

As a whole, the work is conceptually novel for the bone biology community and the newer mechanobiology sub-community within it. The use of *in vivo* and *in vitro* models of mechanical loading/FFSS to study signaling and *Sost* expression is a strength. The work will likely gain attention from a wider field because of the discovery that Hdac5 is phosphorylated by FAK on a tyrosine residue and a new pY-Hdac5 antibody was generated. The manuscript is compelling but additional evidence is needed to strengthen the conclusions.

We thank the reviewer for comments indicating that our work is conceptually novel, compelling, and applicable to a broad audience of investigators studying class IIa HDACs.

1. The conclusions (and title) state that FFSS and FAK control class IIa Hdac signaling. This is too broad as not all class IIa Hdacs were studied. Hdac5 was studied in greatest depth and Hdac4 was assessed in many but not all experiments. There is no mention of Hdac7 or Hdac9 in the text. The RNA-seq data provided in the supplemental table indicate that Hdac7 has similarities to Hdac5 in that it is highly expressed in osteocytes and is down regulated by FFSS and in FAK KO cells. It would be ideal to at least mention potential roles Hdac7 and Hdac9 in the manuscript, perhaps in the Discussion. The title and conclusions should be adjusted as well, focusing on just Hdac4/5 (or perhaps just Hdac5 as the comments below are taken into consideration).

We thank the reviewer for raising this very important point. The majority of our studies here focuses on HDAC5. However, HDAC5 is closely related to HDAC4, and we note that deletion of both HDAC4 and HDAC5 is needed to blunt FFSS-mediated SOST suppression *in vitro*, and only HDAC4/5 compound mutant mice fail to increase periosteal bone formation in our *in vivo* loading studies. Both HDAC5 and HDAC4 translocate into the nucleus in response to FFSS. As noted here by the reviewer, HDAC9 expression is extremely low in Ocy454 cells. For this reason, we have not performed experiments on this class IIa HDAC protein. In contrast, HDAC7

expression is noted in Ocy454 cells. As shown in **Figure R4**, HDAC7 is predominantly cytoplasmic in these cells, and does not translocate into the nucleus in response to FFSS or FAK inhibitor treatment. The discussion of our manuscript has been revised to justify our focus on HDAC5 and, to a lesser degree, HDAC4. This focus is justified based on the clear phenotype seen in cells and mice lacking these two specific class IIa HDACs. As suggested above, the title of the manuscript has been revised to indicate the focus on HDAC5, a gene known to control bone density in mice (11) and humans (12).

Figure R4. (A) RNAseq data from Ocy454 cells grown at 37°C for the indicated times. RPKM counts for the each member of the class IIa HDAC family are shown. (B) Ocy454 cells were treated plus/FFSS for 0 and 60 minutes followed by subcellular fractionation and then immunoblotting.

2. More rationale is needed for the mouse models used in the first two figures and the first two supplemental figures. One issue that needs to be explained is why *Hdac4* cKO mice and *Hdac5* KO mice were used for these studies. The experiments would be more convincing if *Hdac5* cKO mice (*Dmp1-Cre*) were used to show the cell-autonomous role of *Hdac5* in osteoblasts/osteocytes. The use of mixed cKO and KO mice also raises an important issue about controls, which appear to be wildtype and not expressing *Dmp1-Cre*⁺. At the very least, the limitations of these models needs to be addressed.

This is an excellent point. Unfortunately, “floxed” *Hdac5* mutant mice are not available at the present time. Global HDAC5 KO mice were first reported in 2004 (13) and demonstrate minimal basal phenotypes outside of bone. We previously reported that compound *Hdac4* cKO (with *DMP1-Cre*) and global *Hdac5* KO mutants show cortical osteopenia that is more severe than single *Hdac5* mutants (14). Based on the breeding strategies that were used to generate experimental mice for Figure 1, the most appropriate controls to use were *Hdac4*^{fl/fl}; *Hdac5*^{+/+} mice that did not express *Dmp1-Cre*. Multiple groups have demonstrated that expression of the *Dmp1-Cre* transgene on its own has no effect of skeletal traits (for review, (15)). Therefore, we believe that these were the best possible control mice for this experiment. The discussion has been revised to address the limitations mentioned above by the reviewer.

3. Several panels within Figures 1 and 2 and Supplemental Fig 1/2 only show two or three of the four groups needed. For example, only the control and DKO mice are shown in Fig 1B-D. The KO and cKO mice should be added to these figures or to supplemental data. In Figure 2A and supplemental Fig 1 (ABC), the *Hdac4* cKO mice are not shown.

For the immunohistochemistry and SOST bone RNA experiments performed in Figure 1B-D, tissues were only processed from the WT and *Hdac4/5* DKO animals. For Figure 2A, *Hdac4* deficient Ocy454 cells were also analyzed for SOST regulation by FFSS. We note intact FFSS-mediated SOST suppression when HDAC4 alone has been deleted. **New Figure 2A** has been revised to show these results. For Supplemental Figure 1, mechanical testing was only performed on WT, *Hdac5*^{-/-}, and *Hdac4/5* DKO bones. We elected not to perform mechanical testing on the *Hdac4* single mutants because of the absence of a phenotype with respect to cortical bone mass by radiographic and histologic methods. The supplemental figure legend has been revised to clarify this point.

4. Additional details are needed on methods for the loading and FFSS experiments. What is the

age of the mice used for short-term and long-term mechanical loading (p16, line 24)? When were tissues and mRNA isolated? Were they immediately isolated after loading/FFSS stopped or did some time lapse between loading/FFSS and fixation/cell lysis?

We thank the reviewer for providing the opportunity to clarify this point. The methods section has been revised to indicate: "For acute or long-term mechanical loading studies, HDAC5^{-/-};HDAC4^{fl/fl};DMP1-cre female mice and WT littermate controls were subjected to cyclical loading. All mice were subjected to 3 (for acute) or 9 (for long-term) loading episodes on consecutive days, and sacrificed 24 hours after the final round of loading. 20 week old wild type, HDAC4 conditional knockout (cKO, HDAC4 f/f;DMP1-Cre), HDAC5 knockout, and double knockout (DKO, HDAC4 f/f;HDAC5^{-/-};DMP1-Cre) female mice were subjected to *in vivo* cantilever bending of the right tibia. Each mouse underwent a 3 wk regimen (3 d/wk, 100 cycles/d, 2500 μ -epsilon peak normal strain), and dynamic histomorphometry was performed on the tibia mid-shaft."

5. In Figure 2B, the Hdac4 looks mostly cytoplasmic in Ocy454 cells after FFSS, contrary to the conclusions. The densitometric studies on Hdac4 and Hdac5 should be verified with western blots of nuclear and cytoplasmic cell fractions.

As discussed above in response to similar comments by reviewer 2, we have now performed additional experiments using biochemical subcellular fractionation to demonstrate that HDAC5 and (to a lesser extent) HDAC4 translocate from the cytoplasm to the nucleus in response to FFSS. These data are now shown in **new Figure 2D**.

6. The results in Figure 2D are interesting and puzzling. How does Hdac4/5 deficiency prevent Mef2C binding to DNA? Mef2C should be recruiting Hdac4/5 to that region of DNA, not the other way around. Are Mef2C levels in the nucleus different in the DKO cells?

We agree that the results in previous Figure 2D were interesting. Unfortunately, the Mef2c antibody that was used for those ChIP studies is no longer available from Santa Cruz (sc-13266 has been discontinued along with many other products from this company). We tried multiple additional commercially-available Mef2c antibodies for chromatin immunoprecipitation without success. While the mechanism through which HDAC4/5 blocks Mef2c activity remains interesting, this "side question" is outside the main scope of our manuscript. Since the reagents needed for us and others to confirm these results are no longer available, we have decided to remove this figure. Nonetheless, the role of HDAC4/5 in regulating global gene expression changes in response to FFSS remains of high interest. For this reason, we have performed new RNAseq experiments comparing the effects of FFSS in control and HDAC4/5-deficient cells. As shown in **new Figure 2F**, HDAC4/5 DKO cells fail to regulate the majority FFSS-dependent transcripts. These new data add further support to our model that HDAC4/5 play a key role in global gene expression changes in osteocytes in response to mechanical cues.

7. The terms "monoclonal" and "polyclonal" are used on several occasions to distinguish cell lines generated by single-cell cloning from bulk populations transduced with viruses. In immunology, monoclonal is used to describe the unique Ig or TCR sequences formed by DNA recombination. There is no equivalent in osteoblasts/osteocytes. I'm not sure what monoclonal means for virally-transduced cells. Single cell subclones is a better and more traditional description of what was done here.

We thank the reviewer for this thoughtful comment. We used the term "polyclonal" to represent bulk populations of cells transduced with sgRNA-expressing lentiviruses. The term "monoclonal"

was then used to indicate single cell clones derived from these bulk cell populations. Analogous to the physiologic setting on lymphocyte development, these terms (monoclonal and polyclonal) were used to indicate the nature of edited mutations in the distinct cell populations: bulk cells contain many distinct DNA mutations at the targeted site, while single cell clones are 'monoclonal' in that one distinct mutation pattern is present in all cells. This nomenclature is used in many papers describing cells generated by CRISPR/Cas9-mediated genome editing (for example, (16)). That being said, for clarity we have changed "polyclonal" to "bulk", and "monoclonal" to "single cell" for improved clarity.

8. It says that 6 "monoclonal" FAK KO lines were generated (page 20), but it is not clear in the methods what the cell of origin is. Is it Ocy454? Western blots are shown for only 2 in Figure 3B. It appears that 2 clones were used to generate data in Fig 4G too, but this needs clarity. Overall, it is fine to focus on subclones, but there should be rationale for why they were picked, they should be named and used consistently throughout the study.

Supplemental Figure 5B shows FAK western blots from bulk (polyclonal) populations of Cas9-expressing Ocy454 cells transduced with two different FAK (and PYK2) targeting sgRNA lentiviruses (KO1 corresponds to sgRNA-1, and KO2 corresponds to sgRNA-2), as indicated in the figure legend. For Figure 3G (again, please note numbering change to main text figures to accommodate new data), we have clarified the figure legend to indicate that each column reflects RNAseq data from a biologic replicate from a control or FAK-deficient single cell clone. As indicated in the materials/methods section, single cell clones derived from bulk populations were screened for loss of FAK protein by immunoblotting. Clones lacking FAK entirely (derived from initial sgRNA-1 population) were expanded and used for experiments here. As indicated in the revised methods section: "Control cells were transduced with an empty sgRNA-expressing lentivirus." As suggested, the FAK-KO single cell clone used has been named consistently throughout the manuscript as FAK-KO c1-4 (see Figure R5). Of note, in new Supplemental Figure 5C, we now show that defects in SOST expression in this clone are restored upon re-expression of FAK cDNA. When bulk FAK-KO cells were used (supplemental figures only), we refer to these populations as FAK-KO b1 and FAK-KO b2.

Figure R5. (A) Single cell clones derived from bulk FAK sgRNA-infected cells were analyzed for SOST expression by RT-qPCR after 14 days at 37°C. (B) Immunoblot showing FAK protein levels in single cell clones derived bulk FAK sgRNA-infected cells.

9. In Fig 5, were cells treated with FAK inhibitors also LV-transduced and selected? Or were just wildtype cells treated with the inhibitors? Potential effects of LV-transduction on transcriptomic results could be added to an otherwise good discussion on page 9, lines 12-20.

For Figure 4A and 4B, wildtype cells were used to assess acute effects of FAK inhibitors at the biochemical level, as indicated in the figure legend. For Figure 4D-F, we now have clarified the figure legend to note that FAK inhibitor RNAseq was done in wildtype cells, while the FFSS and FAK gene deletion RNAseq was done in LV-transduced cells (FFSS was performed in mock LV-infected cells). We appreciate the opportunity to clarify the important point about effects of LV transduction at the transcriptomic level. The discussion on page 10 (lines 2-7) where RNAseq results are reviewed has been revised: "Finally, we cannot rule out the possibility that some

basal gene expression changes in response to lentiviral infection *per se* may occur. However, all RNAseq studies used cells infected (or not) under identical experimental conditions for side-by-side comparison. Furthermore, principle component analysis of all 28 RNA-seq libraries analyzed here demonstrates only modest effects of lentiviral infection at the transcriptomic level (Supplemental Figure 6B).”

10. To place the results into biological context, it would essential to know what FAK inhibitors do to Ocy454 viability and function at the concentrations tested.

This is an extremely important point. For these studies, we have only treated cells with FAK inhibitors for short times for protein phosphorylation (less than 60 minutes) or gene expression (3-4 hours) endpoints. Based on this comment, we performed additional experiments where we tested the effects of 4 hour treatment of different doses of FAK inhibitors on cell viability using a resavurin-based viability assay. As shown in **new figure 3A**, FAK inhibitors at the doses and treatment times studied here have no effects on Ocy454 cell viability.

11. The role of pY on Hdac4 and Hdac5 nuclear localization deserves more analysis and necessary to support the conclusions (page 11, line 5-7) and models (Fig 8E) proposed. The IF studies in Supp. Fig 4b are insufficient. Examining how pY functions in cooperation with pS would be interesting but not necessary for this study.

We completely agree with this comment, which also was raised by reviewer 2 (point #7). As discussed above, we now show **new Figures 6E-H** indicating that the Hdac5 Y642F mutant shows increased nuclear localization and increased SOST suppression compared to wild type Hdac5.

12. The western blot in Figure 6J indicates that Hdac5 levels were higher in FAK KO and increase further following with FFSS. Hdac5 transcript levels also appear to be higher in the RNA-seq studies. Please discuss the potential meaning of this.

This is an excellent point. It appears that FAK deficiency alters both the overall transcript/protein levels of HDAC5 and its subcellular localization. The primary focus of this work is on the direct regulation of HDAC5 by FAK at the level of Y642 phosphorylation. We have revised the discussion to acknowledge that future studies are needed to understand how FAK deficiency also results in changes in Hdac5 mRNA expression.

13. How does cilengitide affect pY-Hdac4/5 and pS-Hdac4/5, as well as Hdac4 and Hdac5 overall levels? The addition of these western blots to Figure 7BC would link these experiments to the rest of the study better.

This is an important question. Experiments with cilengitide in murine cells are challenging due to species-specific polymorphisms in αV integrin heterodimers (17). For this reason, we have performed analogous experiments in human-derived Saos2 cells, an osteoblastic cell line that produces high amounts of mineralized matrix. As shown in new Figure 7D, cilengitide treatment of (human) Saos2 cells leads to reduction in both FAK and HDAC5 Y642 phosphorylation levels.

14. The FAK inhibitor in vivo experiments in Fig 8 are a reasonable approach in the absence of FAK cKO (Dmp1-cre), but the results in Fig 8a from liver extracts are not convincing.

We thank the reviewer for this important comment. VS-6063 is a well-known FAK inhibitor that is currently under evaluation in clinical trials for mesothelioma, non small cell lung cancer, and other advanced solid tumors (18, 19). In Figure 8A, we have collected liver lysates 4 hours after single dose VS-6063 by intraperitoneal injection. Variability in FAK phosphorylation in these lysates (even from vehicle-treated mice) reflects normal biologic variation in this labile phosphorylation event. Despite this variability, we observe statistically-significant reductions in FAK auto-phosphorylation (as expected) after VS-6063 administration. Along with this, we observe statistically-significant reductions in SOST expression in bone RNA in these mice. For both parameters measured (pFAK in liver and SOST mRNA in bone), variation in vehicle-treated mice likely reflects a combination of biologic and technical variation.

15. In the Discussion, lines 16-17 on page 13 contradict what is stated in lines 1-2 of page 14. The latter is more accurate unless experiments requested above on how pY affects Hdac4/5 nuclear localization can be provided.

We now show new data (see point #11 above) providing additional information about the role of HDAC5 Y642 phosphorylation. As currently worded, this part of the discussion (now on page 14, lines 11-22) indicates two distinct points:

1. That HDAC5 Y642 phosphorylation is important in controlling the subcellular localization of HDAC5
2. That the precise role of HDAC5 Y642 phosphorylation in controlling the subcellular localization of HDAC5 remains to be determined.

Our data clearly supports an important function for HDAC5 Y642 phosphorylation; however, we acknowledge that we do not fully understand how HDAC5 Y642 phosphorylation participates in cytoplasmic retention of this factor. Therefore, we believe that the two statements mentioned above are not necessarily mutually exclusive.

Minor comments:

1. On page 2, line 5, HDACs are called "sensors". This was not shown. Better terms would be transducers, facilitators or amplifiers of mechanical signals and cell membrane interactions.

We agree. As indicated above (Reviewer 2, Point 3), the term sensor has been changed to transducer.

2. In Figure 1A, Hdac4 cKO is abbreviated H4KO in the graphs. This is misleading and should be changed to Hdac4 cKO. Ditto for Supplemental figure 2A.

We agree. For our in vivo studies, the term H4KO has been changed to H4 cKO to reflect that this mutant strain is a conditional knockout of *Hdac4* using the DMP1-Cre transgene.

3. Please clarify which cells were transfected with Hdac5 expression vectors in the legend of supplemental figure 2b. The methods section indicates that it was the Hdac5 KO cells, but Ocy454 cells are mentioned in the legend for panel C so it is a bit confusing as to what is shown in Suppl Fig 2b.

We thank the reviewer for providing the opportunity to clarify this supplemental figure legend. We have revised the figure legend to indicate that Supplemental Figure 3B and 3C were performed in HDAC5-deficient cells reconstituted with wild type of 2SA variants of HDAC5.

4. Page 7, line 2: Please clarify what "rapid" describes or omit this word. 3 hours doesn't seem "rapid" for signaling studies.

This is also an excellent point. We have removed the word "rapid" and now the text simply states that this was a 3 hour study for gene expression purposes.

5. It would be helpful to define what is meant by the term "tonic" signaling when the term is used for the first time. It may be unfamiliar to some in the bone community. It appears to refer to constitutive signaling/kinase activity in the basal state.

We agree. The word "tonic" has been changed to "constitutive" to more clearly describe our model in which constitutive interactions between osteocytes and their surrounding extracellular matrix activates FAK at steady state.

Sincerely,

Marc Wein, MD/PhD

References

1. Prasad J, Wiater BP, Nork SE, Bain SD, and Gross TS. Characterizing gait induced normal strains in a murine tibia cortical bone defect model. *Journal of biomechanics*. 2010;43(14):2765-70.
2. Srinivasan S, Balsiger D, Huber P, Ausk BJ, Bain SD, Gardiner EM, et al. Static Preload Inhibits Loading-Induced Bone Formation. *JBMR Plus*. 2019;3(5):e10087.
3. Sugiyama T, Price JS, and Lanyon LE. Functional adaptation to mechanical loading in both cortical and cancellous bone is controlled locally and is confined to the loaded bones. *Bone*. 2010;46(2):314-21.
4. Lynch ME, Main RP, Xu Q, Schmicker TL, Schaffler MB, Wright TM, et al. Tibial compression is anabolic in the adult mouse skeleton despite reduced responsiveness with aging. *Bone*. 2011;49(3):439-46.
5. Srinivasan S, Weimer DA, Agans SC, Bain SD, and Gross TS. Low-magnitude mechanical loading becomes osteogenic when rest is inserted between each load cycle. *Journal of bone and mineral research : the official journal of the American Society for Bone and Mineral Research*. 2002;17(9):1613-20.
6. Spatz JM, Wein MN, Gooi JH, Qu Y, Garr JL, Liu S, et al. The Wnt Inhibitor Sclerostin Is Up-regulated by Mechanical Unloading in Osteocytes in Vitro. *J Biol Chem*. 2015;290(27):16744-58.
7. Baron R, and Kneissel M. WNT signaling in bone homeostasis and disease: from human mutations to treatments. *Nature medicine*. 2013;19(2):179-92.
8. Tu X, Rhee Y, Condon KW, Bivi N, Allen MR, Dwyer D, et al. Sost downregulation and local Wnt signaling are required for the osteogenic response to mechanical loading. *Bone*. 2012;50(1):209-17.
9. Kang KS, Hong JM, and Robling AG. Postnatal beta-catenin deletion from Dmp1-expressing osteocytes/osteoblasts reduces structural adaptation to loading, but not periosteal load-induced bone formation. *Bone*. 2016;88:138-45.
10. Gelens L, and Saurin AT. Exploring the Function of Dynamic Phosphorylation-Dephosphorylation Cycles. *Developmental cell*. 2018;44(6):659-63.

11. Wein MN, Spatz J, Nishimori S, Doench J, Root D, Babij P, et al. HDAC5 controls MEF2C-driven sclerostin expression in osteocytes. *Journal of bone and mineral research : the official journal of the American Society for Bone and Mineral Research*. 2015;30(3):400-11.
12. Rivadeneira F, Styrkarsdottir U, Estrada K, Halldorsson BV, Hsu YH, Richards JB, et al. Twenty bone-mineral-density loci identified by large-scale meta-analysis of genome-wide association studies. *Nature genetics*. 2009;41(11):1199-206.
13. Chang S, McKinsey TA, Zhang CL, Richardson JA, Hill JA, and Olson EN. Histone deacetylases 5 and 9 govern responsiveness of the heart to a subset of stress signals and play redundant roles in heart development. *Mol Cell Biol*. 2004;24(19):8467-76.
14. Wein MN, Liang Y, Goransson O, Sundberg TB, Wang J, Williams EA, et al. SIKs control osteocyte responses to parathyroid hormone. *Nature communications*. 2016;7:13176.
15. Dallas SL, Xie Y, Shiflett LA, and Ueki Y. Mouse Cre Models for the Study of Bone Diseases. *Current osteoporosis reports*. 2018;16(4):466-77.
16. Yang L, Yang JL, Byrne S, Pan J, and Church GM. CRISPR/Cas9-Directed Genome Editing of Cultured Cells. *Curr Protoc Mol Biol*. 2014;107:31 1 1-17.
17. Blue R, Kowalska MA, Hirsch J, Murcia M, Janczak CA, Harrington A, et al. Structural and therapeutic insights from the species specificity and in vivo antithrombotic activity of a novel alphaIIb-specific alphaIIb beta3 antagonist. *Blood*. 2009;114(1):195-201.
18. Fennell DA, Baas P, Taylor P, Nowak AK, Gilligan D, Nakano T, et al. Maintenance Defactinib Versus Placebo After First-Line Chemotherapy in Patients With Merlin-Stratified Pleural Mesothelioma: COMMAND-A Double-Blind, Randomized, Phase II Study. *J Clin Oncol*. 2019;37(10):790-8.
19. Gerber DE, Camidge DR, Morgensztern D, Cetnar J, Kelly RJ, Ramalingam SS, et al. Phase 2 study of the focal adhesion kinase inhibitor defactinib (VS-6063) in previously treated advanced KRAS mutant non-small cell lung cancer. *Lung Cancer*. 2020;139:60-7.

b>REVIEWER COMMENTS

Reviewer #1 (Remarks to the Author):

I had in the previous version serious issues with the mechanical loading used and asked to explain it more in depth and verify what happened in vivo to be able to draw and of the in depth conclusions on molecular cues that the authors intended to do:

After this review, the loading remains to be unclear – what the authors define as “cantilever” and being physiological to gait is hard to believe: There is no drawing or image on how the bones have been loaded. The paper you reference has discussed bone defect strains and used an inverse dynamics approach to postulate tissue strains in a purely in silico approach. None of the published is directly related to the strains the authors have had in their cantilever bending of the tibia in vivo. With the lack of information it is hard to believe that the tissue strain was (a) homogenous across a cross section (looks like quite a bit of shear on top of bending and thus being NOT homogenous), (b) reproducible with small variations altering fundamental the tissue straining and (c) comparable to physiological situations since neither mice nor humans walk with a potted knee joint!

Both references given by the authors in the rebuttal (but not in the manuscript: “Our previous studies have demonstrated that the mid-shaft normal strain distribution induced by our model of cantilever bending (not 3-pt bending) closely mirrors that induced by locomotion (1, 2)”) are not referring to in vivo data. In vivo tissue straining has not been measured in the present model and thus it cannot be benchmarked against in vivo physiological data or any other limb loading data. The claims that the strain is homogenous or physiological have never been verified, not in the present paper not in the referenced papers.

Since no details are given, assume the mechanics are not controlled and thus any further discussion of more sophisticated details does not really make any sense... This manuscript would not survive a review in any biomechanical journal!

Authors claim (“This adaptive response is reminiscent of the modest periosteal bone formation induced in humans by exercise interventions and is consistent with current bone mechanotransduction paradigms (i.e., osteocyte mechanosensing translated into a periosteal bone formation response)”) but give no or minimal details on the loading system such that a reproduction of the analyses is not possible. I would thus reject the paper.

The authors statement that the result verify the model (“Our in vivo results (downregulation of osteocyte sclerostin in loaded WT, but loss of this effect in Hdac4/5 DKO, in parallel with the loss of load induced periosteal osteoblastic response, Fig 1a and b) support the experimental approach used here.”) are not really helpful to introduce a novel loading idea. If the loading model is inconsistent (no consistent loading across the crosssection, the findings on the periosteal bone solely a result from the model) any details derived thereafter are falling short on having any physiological relevance. On top, authors claim to have included fluid flow – which I have not seen any evidence for. These were purely looking at tissue straining and also not explaining the loading protocol in enough detail AND none of this has been verified in vivo. References are all pure simulation with NO in vivo tissue strain measurements.

On my request to be “More Specific: A figure illustrating the loading and tissue sampling should be included to estimate/illustrate the resulting tissue straining” a Figure 1 in the supplements was added that is simply a copy from the FEA work published in J Biomechanics some years ago(!). What is needed is (a) a figure of how mice were loaded (b) what tissue strains result from this – here the current image might help and (c) an in vivo strain analyses at one spot on the cortex to verify that the simulation hold at least roughly true. All this basic characterization of a qualified in vivo (artificial) loading set-up is missing but claims are made that it matches physiology...

My question "The manuscript currently mixes the mechanical stimulus that comes from limb loading and the one that comes from fluid flow and leads to control of osteocyte mechanotransduction. These mechanical stimuli, however, are different. The paper presents in vitro and in vivo data on in depth mechanistic analyses of mechanotransduction to proof this claim but mixes these mechanical stimuli and their effects on tissues (limb loading leads to osteoblastic and osteoclastic activity at the endost or periost bone surface; fluid flow around osteocytes would lead to alterations in bone mineral within the cortex). Please clarify which effect you concentrate on." Have not been answered. It remains unclear what mechanical stimulation was effective and how homogeneous it was and if it was mainly addressing Osteoblasts, osteoclasts, endosteal or periosteal bone or osteocytes via fluid flow – complete different process. All is mixed...

Reviewer #2 (Remarks to the Author):

The authors' additional experimental data (including biochemical cell fractionation to support nuclear localization and reconstitution of the CRISPR FAK knockout cells, and functional analysis of the HDAC5 tyrosine to phenylalanine point mutation) greatly enhances the impact of this study.

Please consider these additional wording changes:

1) Note that FAK activity parallels changes in FAK Y397 phosphorylation, but it is only an indirect measure.

In the Abstract line 31 should be changed to:

"Osteocyte cell adhesion supports FAK tyrosine phosphorylation and FFSS rapidly triggers FAK dephosphorylation. Pharmacologic FAK catalytic inhibition reduces osteocyte Sost mRNA expression in vitro and in vivo."

2) The experiments with celinegite (a specific inhibitor of $\alpha_5\beta_3$ integrins) are problematic with respect to linkages to collagen binding in bones. These integrins don't bind collagen. Additionally, use of RGD peptide addition is a cellular "hammer". For human cells, there are specific integrin blocking antibodies. This part of the study remains a bit weak. I would suggest removal as it does not impact the main findings (and could be part of future studies).

3) The notion of "constitutive" FAK activity in the basal (in vitro) cell conditions is a bit difficult to reconcile. As studies to date have linked INCREASED FAK tyrosine phosphorylation with FFSS (and the authors are reporting an opposite result with the osteocytes - FFSS resulting in DECREASED FAK tyrosine phosphorylation), word choice is very important. This reviewer would suggest that the authors use the descriptor "basal FAK signaling" to denote the signals supporting HDAC5 phosphorylation and changes in Sost mRNA levels prior to FFSS. There are multiple places in the manuscript where this needs to be addressed.

Reviewer #3 (Remarks to the Author):

Thank you for addressing my concerns. The only minor recommendation is to use the same font/font size in all figures. A readable font like Helvetica or Arial is preferred. The beta (beta-actin) in y-axis labels is distorted and should be fixed too.

Nice work!

Reviewer #4 (Remarks to the Author):

I have been asked to comment on Reviewer #1's "issues with the mechanical loading used." R1's concerns center exclusively upon the in vivo loading experiments. These demonstrate that application of a cantilevered loading paradigm in mice results in decreased sclerostin expression (1B) in the loaded bone as well as periosteal bone formation (Fig1A); in where HDACs 4 and 5 are knocked out in osteocytes (global Hdac5 null x DMP1-cre; Hdac4 null), this does not occur. This outstanding data is a firm starting point for the in vitro mechanistic studies that follow.

The in vivo data for this paper convincingly introduces the in vitro experiments that makes up the bulk of the paper. The cantilever loading model, shown in the supplementary data (as requested by R1) is well accepted in the field and by bioengineers. As this is not a novel loading paradigm, it does not require a reassessment as it was reviewed by bioengineers when originally published (reference #98 in the version provided me), as well as in many other publications. Finally, insights initially identified with this model have been subsequently confirmed in other models (e.g., Grimston PLoS One 2012 PMID: 22970183, Bivi J Orthop Res 2013 PMID:23483620).

As to the homogeneity in the animal bending strain model: tissue strain is heterogenous! Indeed Dr. Gross, the biomechanics expert co-authoring this paper, has published extensively on this subject, and his conclusions that the tissue (bone) responds as a complex system (Gross JBMR 1997 PMID:9169359) are widely accepted. It is worth noting that strains are almost always calculated rather than measured with strain gages, as in silico methods were verified decades ago. Thus, is it puzzling and outside of this scope of this manuscript that R1 asks for exacting data as to tissue strain.

The in vitro studies that follow (Fig 2 on) utilize fluid flow shear strain (FFSS), rather than strain due to load. It is well known that strain generates fluid shear, and that heterogenous strain leads to fluid flow around osteocytes (references #28,29). In vivo, it is not possible to have one without the other. In vitro studies generally refer to one or the other, but it is highly likely that 8 dynes/cm FFSS generates strain. The physiologic summation of strain, pressure and shear flow that occur with in vivo loading is not meant to be recreated in vitro (because it can't!). As such, a signaling study represents, at the best, a thoughtful model of a complex and immeasurably temporal physiology. The reductionism required to perform mechanistic studies is accepted by (most) scientists as a necessary condition to advance fundamental understanding as well as lead to clinical strategies to treat pathological conditions.

I strongly recommend that R1's arrogations with regard to the worthiness of the extensive, careful and novel work in this manuscript be dismissed. Not only would the leading in vivo data in Fig 1 stand up to review in a biomechanics journal, the work leads where readers want to follow – into novel mechanisms.

We thank the reviewers for their valuable input. Our point-by-point responses (in red) to the specific concerns raised by each reviewer (in black) are found below. In the revised manuscript text, all changes are highlighted in yellow.

Reviewer #1 (Remarks to the Author):

General Response

Overall, R1 felt that we were not sufficiently detailed in our response to the issues that were raised. However, our responses were framed from our understanding that the manuscript is neither a skeletal loading study nor describes a novel loading device. Rather, the *in vivo* loading data were explored to provide physiologic context for the detailed mechanotransduction signaling studies that constitute the primary focus of the manuscript. Thus, as our responses to the initial skeletal loading queries were intended to be informational for the reviewers, we revised the manuscript with a focus toward improving the manuscript for a general audience.

In this context, we addressed two general themes arising from R1 comments. First, the reviewer frequently refers to our device as a novel loading model. However, this model is not novel (per Pubmed, at a minimum, 11 original studies (primarily in journals publishing bone adaptation studies), 2 reviews, and 2 in vitro studies have been based on data generated from this model since its initial publication in 2002). Further, the strategy we used to equilibrate peak normal strains across groups is the same approach we have applied in a number of similar studies (1, 2).

Second, R1 infers that if the strains induced by the external loading device are not defined in great detail, all of the *in vivo* data should be disregarded. We believe that two observations suggest that this inference is inappropriate: 1) the field standard is to present detailed calibrations only when a new loading device is published for the first time (e.g., as in our original paper with the model used in the current study (3), and a new model we have recently developed (4)), and 2) the end loading applied to the tibia was controlled so that peak induced normal strains at the mid-shaft were equivalent across groups (a strategy we have repeatedly used in previous studies). This objective is achieved via combined FEA and in situ strain gaging of the induced normal strain environment.

I had in the previous version serious issues with the mechanical loading used and asked to explain it more in depth and verify what happened in vivo to be able to draw and of the in depth conclusions on molecular cues that the authors intended to do:

After this review, the loading remains to be unclear – what the authors define as “cantilever” and being physiological to gait is hard to believe: There is no drawing or image on how the bones have been loaded. The paper you reference has discussed bone defect strains and used an inverse dynamics approach to postulate tissue strains in a purely in silico approach. None of the published is directly related to the strains the authors have had in their cantilever bending of the tibia in vivo. With the lack of information it is hard to believe that the tissue strain was (a) homogenous across a cross section (looks like quite a bit of shear on top of bending and thus being NOT homogenous), (b) reproducible with small variations altering fundamental the tissue straining and (c) comparable to physiological situations since neither mice nor humans walk with a potted knee joint!

1. A detailed schematic of the cantilever bending model can be found in the manuscript that originally described the model (and referenced in Supplemental Figure 1). The FEA analysis, which is standard for the field, demonstrates that the strains within a given cross-section are heterogeneous, as one would expect for bending (tension on one side, compression on the other, please see **revised Supplemental Fig 1**). Since we first reported the model in 2002, we have performed nearly 40 FEA/strain gage calibrations for a variety of studies with this device and have loaded over 1500 mice. Many of these studies required us to adjust applied external loads across experimental groups (due to genetic- or age- related morphologic differences) in order reproducibly equilibrate peak normal strains across the groups. Finally, we have clarified the manuscript to indicate that the cantilever bending device does not model dynamic loading of the tibia induced during gait, but, rather, at the tibia mid-shaft the neutral axis by the loading device is similar in orientation to that induced by gait (pg 38).

Both references given by the authors in the rebuttal (but not in the manuscript: “Our previous studies have demonstrated that the mid-shaft normal strain distribution induced by our model of cantilever bending (not 3-pt bending) closely mirrors that induced by locomotion (1, 2)”) are not referring to in vivo data. In vivo tissue straining has not been measured in the present model and thus it cannot be benchmarked against in vivo physiological data or any other limb loading data. The claims that the strain is homogenous or physiological have never been verified, not in the present paper not in the referenced papers.

2. Thank you, we have now included these references as requested.

Since no details are given, assume the mechanics are not controlled and thus any further discussion of more sophisticated details does not really make any sense... This manuscript would not survive a review in any biomechanical journal!

3. Please see our general response above. For this manuscript, we utilized an established model that has been rigorously calibrated in numerous previous studies.

Authors claim (“This adaptive response is reminiscent of the modest periosteal bone formation induced in humans by exercise interventions and is consistent with current bone mechanotransduction paradigms (i.e., osteocyte mechanosensing translated into a periosteal bone formation response)”) but give no or minimal details on the loading system such that a reproduction of the analyses is not possible. I would thus reject the paper.

4. As the loading device and protocol has been implemented in previous studies, we have **expanded the methods section and citations** that would enable reproduction of the study design by other colleagues (pg 19).

The authors statement that the result verify the model (“Our *in vivo* results (downregulation of osteocyte sclerostin in loaded WT, but loss of this effect in Hdac4/5 DKO, in parallel with the loss of load induced periosteal osteoblastic response, Fig 1a and b) support the experimental approach used here.”) are not really helpful to introduce a novel loading idea. If the loading model is inconsistent (no consistent loading across the crosssection, the findings on the periosteal bone solely a result from the model) any details derived thereafter are falling short on having any physiological relevance. On top, authors claim to have included fluid flow – which I have not seen any evidence for. These were purely looking at tissue straining and also not explaining the loading protocol in enough detail AND none of this has been verified *in vivo*. References are all pure simulation with NO *in vivo* tissue strain measurements.

5. Please see our general response and specific response #1 above.

On my request to be “More Specific: A figure illustrating the loading and tissue sampling should be included to estimate/illustrate the resulting tissue straining” a Figure 1 in the supplements was added that is simply a copy from the FEA work published in J Biomechanics some years ago(!). What is needed is (a) a figure of how mice were loaded (b) what tissue strains result from this – here the current image might help and (c) an *in vivo* strain analyses at one spot on the cortex to verify that the simulation hold at least roughly true. All this basic characterization of a qualified *in vivo* (artificial) loading set-up is missing but claims are made that it matches physiology...

6. We have addressed these concerns in our general response above and have referenced the paper where the loading device was initially described. We have **revised Supplemental Figure 1** to include the schematic from the initial loading device study as it clarifies placement of the mouse in the loading device.

My question “The manuscript currently mixes the mechanical stimulus that comes from limb loading and the one that comes from fluid flow and leads to control of osteocyte mechanotransduction. These mechanical stimuli, however, are different. The paper presents *in vitro* and *in vivo* data on in depth mechanistic analyses of mechanotransduction to proof this claim but mixes these mechanical stimuli and their effects on tissues (limb loading leads to osteoblastic and osteoclastic activity at the endost or periost bone surface; fluid flow around osteocytes would lead to alterations in bone mineral within the cortex). Please clarify which effect you concentrate on.” Have not been answered. It remains unclear what mechanical stimulation was effective and how homogenous it was and if it was mainly addressing Osteoblasts, osteoclasts, endosteal or periosteal bone or osteocytes via fluid flow – complete different process. All is mixed...

7. As noted above, the purpose of the *in vivo* loading experiment was to assess whether HDAC4/5 deficiency altered the anabolic response of bone to controlled skeletal loading. These data provided context for the subsequent mechanistic *in vitro* studies. Further, as Reviewer #4 noted, the manuscript does not attempt to explore which direct (e.g., bone deformation) or indirect (e.g., fluid flow) stimuli are primarily responsible for mediating bone adaptation to skeletal loading. Given the state of the art (and as Reviewer #4 also observed), it is not possible to decouple these inter-related stimuli *in vivo*. **We have revised the discussion** to reflect this complexity (pg 15, line 3-7).

Reviewer #2 (Remarks to the Author):

The authors additional experimental data (including biochemical cell fractionation to support nuclear localization and reconstitution of the CRISPR FAK knockout cells, and functional analysis of the HDAC5 tyrosine to phenylalanine point mutation) greatly enhances the impact of this study.

Please consider these additional wording changes:

1) Note that FAK activity parallels changes in FAK Y397 phosphorylation, but it is only an indirect measure.

In the Abstract line 31 should be changed to:

"Osteocyte cell adhesion supports FAK tyrosine phosphorylation and FFSS rapidly triggers FAK dephosphorylation. Pharmacologic FAK catalytic inhibition reduces osteocyte Sost mRNA expression in vitro and in vivo."

We thank the reviewer for this suggestion. We completely agree that FAK Y397 phosphorylation is used as a surrogate for FAK activity. The abstract has been revised accordingly.

2) The experiments with cilingitide (a specific inhibitor of $\alpha v\beta 3$ integrins) are problematic with respect to linkages to collagen binding in bones. These integrins don't bind collagen. Additionally, use of RGD peptide addition is a cellular "hammer". For human cells, there are specific integrin blocking antibodies. This part of the study remains a bit weak. I would suggest removal as it does not impact the main findings (and could be part of future studies).

This is an excellent point. In addition to inhibiting $\alpha v\beta 3$ integrins, cilengitide also inhibits $\alpha 5\beta 1$ heterodimers. Bone matrix is rich in non-collagenous ECM components such as osteopontin and fibronectin, both of which are abundantly expressed in Ocy454 cells. Osteopontin and fibronectin both bind $\alpha v\beta 3$ and $\alpha 5\beta 1$ integrin heterodimers. We agree that future work is needed to definitively identify the ECM component/integrin pair that drives high basal FAK activity in osteocytes. That being said, the studies with cilengitide (strengthened during the revision process by new data with human Saos2 cells) contribute to our current model. Based on this important comment, the Discussion (page 16, paragraph starting on line 6) has been revised accordingly.

3) The notion of "constitutive" FAK activity in the basal (in vitro) cell conditions is a bit difficult to reconcile. As studies to date have linked INCREASED FAK tyrosine phosphorylation with FFSS (and the authors are reporting an opposite result with the osteocytes - FFSS resulting in DECREASED FAK tyrosine phosphorylation), word choice is very important. This reviewer would suggest that the authors use the descriptor "basal FAK signaling" to denote the signals supporting HDAC5 phosphorylation and changes in Sost mRNA levels prior to FFSS. There are multiple places in the manuscript where this needs to be addressed.

We thank the reviewer for this important comment. The surprising observation that FFSS reduces FAK tyrosine phosphorylation in Ocy454 cells is first addressed on page 12 in the paragraph starting on line 11. Throughout the manuscript, we have followed the reviewer's excellent suggestion and used the descriptor 'basal FAK signaling'.

Reviewer #3 (Remarks to the Author):

Thank you for addressing my concerns. The only minor recommendation is to use the same font/font size in all figures. A readable font like Helvetica or Ariel is preferred. The beta (beta-

actin) in y-axis labels is distorted and should be fixed too.

Nice work!

We agree- the figures have been revised for consistency and readability.

Reviewer #4 (Remarks to the Author):

I have been asked to comment on Reviewer #1's "issues with the mechanical loading used." R1's concerns center exclusively upon the in vivo loading experiments. These demonstrate that application of a cantilevered loading paradigm in mice results in decreased sclerostin expression (1B) in the loaded bone as well as periosteal bone formation (Fig1A); in where HDACs 4 and 5 are knocked out in osteocytes (global Hdac5 null x DMP1-cre; Hdac4 null), this does not occur. This outstanding data is a firm starting point for the in vitro mechanistic studies that follow.

The in vivo data for this paper convincingly introduces the in vitro experiments that makes up the bulk of the paper. The cantilever loading model, shown in the supplementary data (as requested by R1) is well accepted in the field and by bioengineers. As this is not a novel loading paradigm, it does not require a reassessment as it was reviewed by bioengineers when originally published (reference #98 in the version provided me), as well as in many other publications. Finally, insights initially identified with this model have been subsequently confirmed in other models (e.g., Grimston PLoS One 2012 PMID: 22970183, Bivi J Orthop Res 2013 PMID:23483620).

As to the homogeneity in the animal bending strain model: tissue strain is heterogenous! Indeed Dr. Gross, the biomechanics expert co-authoring this paper, has published extensively on this subject, and his conclusions that the tissue (bone) responds as a complex system (Gross JBMR 1997 PMID:9169359) are widely accepted. It is worth noting that strains are almost always calculated rather than measured with strain gages, as in silico methods were verified decades ago. Thus, is it puzzling and outside of this scope of this manuscript that R1 asks for exacting data as to tissue strain.

The in vitro studies that follow (Fig 2 on) utilize fluid flow shear strain (FFSS), rather than strain due to load. It is well known that strain generates fluid shear, and that heterogenous strain leads to fluid flow around osteocytes (references #28,29). In vivo, it is not possible to have one without the other. In vitro studies generally refer to one or the other, but it is highly likely that 8 dynes/cm FFSS generates strain. The physiologic summation of strain, pressure and shear flow that occur with in vivo loading is not meant to be recreated in vitro (because it can't!). As such, a signaling study represents, at the best, a thoughtful model of a complex and immeasurably temporal physiology. The reductionism required to perform mechanistic studies is accepted by (most) scientists as a necessary condition to advance fundamental understanding as well as lead to clinical strategies to treat pathological conditions.

I strongly recommend that R1's arrogations with regard to the worthiness of the extensive, careful and novel work in this manuscript be dismissed. Not only would the leading in vivo data in Fig 1 stand up to review in a biomechanics journal, the work leads where readers want to follow – into novel mechanisms.

We thank the reviewer for these insightful and supportive comments. We agree that the *in vivo* studies performed in Figure 1 use a well-established *in vivo* loading model which sets the stage for the detailed signaling studies performed in the subsequent figures.

Sincerely,

Marc Wein, MD/PhD

References

1. Zhang Y, Paul EM, Sathyendra V, Davison A, Sharkey N, Bronson S, et al. Enhanced osteoclastic resorption and responsiveness to mechanical load in gap junction deficient bone. *PloS one*. 2011;6(8):e23516.
2. Hankenson KD, Ausk BJ, Bain SD, Bornstein P, Gross TS, and Srinivasan S. Mice lacking thrombospondin 2 show an atypical pattern of endocortical and periosteal bone formation in response to mechanical loading. *Bone*. 2006;38(3):310-6.
3. Gross TS, Srinivasan S, Liu CC, Clemens TL, and Bain SD. Noninvasive loading of the murine tibia: an in vivo model for the study of mechanotransduction. *Journal of bone and mineral research : the official journal of the American Society for Bone and Mineral Research*. 2002;17(3):493-501.
4. Srinivasan S, Balsiger D, Huber P, Ausk BJ, Bain SD, Gardiner EM, et al. Static Preload Inhibits Loading-Induced Bone Formation. *JBMR Plus*. 2019;3(5):e10087.

REVIEWERS' COMMENTS

Reviewer #1 (Remarks to the Author):

Dear authors of the manuscript "A FAK/HDAC5 signaling axis controls osteocyte mechanotransduction" (Tadatoshi Sato et al.), thank you very much for the clarifications you added. Extending the loading device description really helps readers to digest the approach and I am sorry that my insisting produced so much work for you. However, I feel it was worth it.

May I ask for two small details: In the discussion you write "Though many aspects of mechanical loading (including direct tissue strain and secondary interstitial fluid flow) are known to be anabolic to bone cells, it is not possible to decouple these stimuli in vivo. However, in this context, the heterogeneous fluid flow stimuli induced via external skeletal loading (68-70) is one stimulus known to activate both osteocytes and differentially activate bone surface cells (71)." Maybe adding a statement here that the link between osteocyte mechano-sensation along the cascades you describe and the details of osteoblastic bone formation remains, however, to be unraveled?

Second, figure 1 shows very nicely the SOST positive OC in WT and H4H5-DKO mice after loading. Does the location of the positive /resp. negative cells match in WT the strain fields in this bone plane (roughly)? A hint to such match - maybe I overlooked it - would be a nice add to the results/discussion and further strengthen the key findings.

Reviewer #4 (Remarks to the Author):

Reviewer 4 has no further comments.

Our point-by-point responses (in red) to the concerns raised by each reviewer (in black) are below. All changes are highlighted in yellow in the revised manuscript.

Reviewer #1 (Remarks to the Author):

Dear authors of the manuscript "A FAK/HDAC5 signaling axis controls osteocyte mechanotransduction" (Tadatoshi Sato et al.), thank you very much for the clarifications you added. Extending the loading device description really helps readers to digest the approach and I am sorry that my insisting produced so much work for you. However, I feel it was worth it. May I ask for two small details: In the discussion you write "Though many aspects of mechanical loading (including direct tissue strain and secondary interstitial fluid flow) are known to be anabolic to bone cells, it is not possible to decouple these stimuli *in vivo*. However, in this context, the heterogeneous fluid flow stimuli induced via external skeletal loading (68-70) is one stimulus known to activate both osteocytes and differentially activate bone surface cells (71)." Maybe adding a statement here that the link between osteocyte mechano-sensation along the cascades you describe and the details of osteoblastic bone formation remains, however, to be unraveled?

We agree, and thank the reviewer for this important question. The discussion has now been revised with the statement "The precise link between the osteocyte mechano-transduction signaling cascade described here and skeletal responses to loading *in vivo* remains to be unraveled" on page 15 (lines 7-9).

Second, figure 1 shows very nicely the SOST positive OC in WT and H4H5-DKO mice after loading. Does the location of the positive /resp. negative cells match in WT the strain fields in this bone plane (roughly)? A hint to such match - maybe I overlooked it - would be a nice add to the results/discussion and further strengthen the key findings.

This is an excellent question. Other investigators have explored the relationship between predicted patterns of strain and sclerostin down-regulation at the level of individual osteocytes in distinct locations in cortical bone. We have revised page 6 (lines 1-2) to include reference 46 which is one of the best studies that addresses this point.

Sincerely,

Marc Wein, MD/PhD